# L-WISE: Boosting human visual category learning through model-based image selection and enhancement

**Morgan B. Talbot**[1,2,3]**, Gabriel Kreiman**[1,2]**, James J. DiCarlo**[2,4,5]**, and Guy Gaziv**[2,4]
[1]Boston Children's Hospital, Harvard Medical School
[2]Center for Brains, Minds, and Machines, MIT
[3]Dept. of Health Sciences and Technology, MIT
[4]McGovern Institute for Brain Research, Dept. of Brain and Cognitive Sciences, MIT
[5]MIT Quest for Intelligence

## Abstract

The currently leading artificial neural network models of the visual ventral stream – which are derived from a combination of performance optimization and robustification methods – have demonstrated a remarkable degree of behavioral alignment with humans on visual categorization tasks. We show that image perturbations generated by these models can enhance the ability of humans to accurately report the ground truth class. Furthermore, we find that the same models can also be used out-of-the-box to predict the proportion of correct human responses to individual images, providing a simple, human-aligned estimator of the relative difficulty of each image. Motivated by these observations, we propose to augment visual learning in humans in a way that improves human categorization accuracy at test time. Our learning augmentation approach consists of (i) selecting images based on their model-estimated recognition difficulty, and (ii) applying image perturbations that aid recognition for novice learners. We find that combining these model-based strategies leads to categorization accuracy gains of 33-72% relative to control subjects without these interventions, on unmodified, randomly selected held-out test images. Beyond the accuracy gain, the training time for the augmented learning group was also shortened by 20-23%, despite both groups completing the same number of training trials. We demonstrate the efficacy of our approach in a fine-grained categorization task with natural images, as well as two tasks in clinically relevant image domains – histology and dermoscopy – where visual learning is notoriously challenging. To the best of our knowledge, our work is the first application of artificial neural networks to increase visual learning performance in humans by enhancing category-specific image features.

*Code*    *Webpage*

## 1 Introduction

Over the last decade, artificial neural network (ANN) models have demonstrated superior performance as image-computable emulators of neural processing along the human and monkey ventral visual stream. Iterative efforts have developed models that are increasingly aligned with primate vision, as measured by their ability to predict both neural activity and behavioral responses (Schrimpf et al., 2018). Beyond prediction, a promising class of these models – "robustified" deep ANNs (Mądry et al., 2018) – has been shown to enable the generation of image perturbations that predictably control both ventral stream neural activity (Guo et al., 2022) and human object categorization reports (Gaziv et al., 2023; Croce & Hein, 2020). In our work, we ask whether the prediction and control capabilities of robustified ANNs can be used to enhance human performance at visual categorization tasks.

Beyond categorization of familiar visual categories, one practically important task is learning to recognize new, unfamiliar categories. Although humans can readily learn many new categories even from a single example (Lake et al., 2011), some consequential tasks require extensive training to reach high levels of performance. For example, medical specialists such as pathologists and radiologists devote numerous hours to mastering the diagnosis of various diseases from medical images.

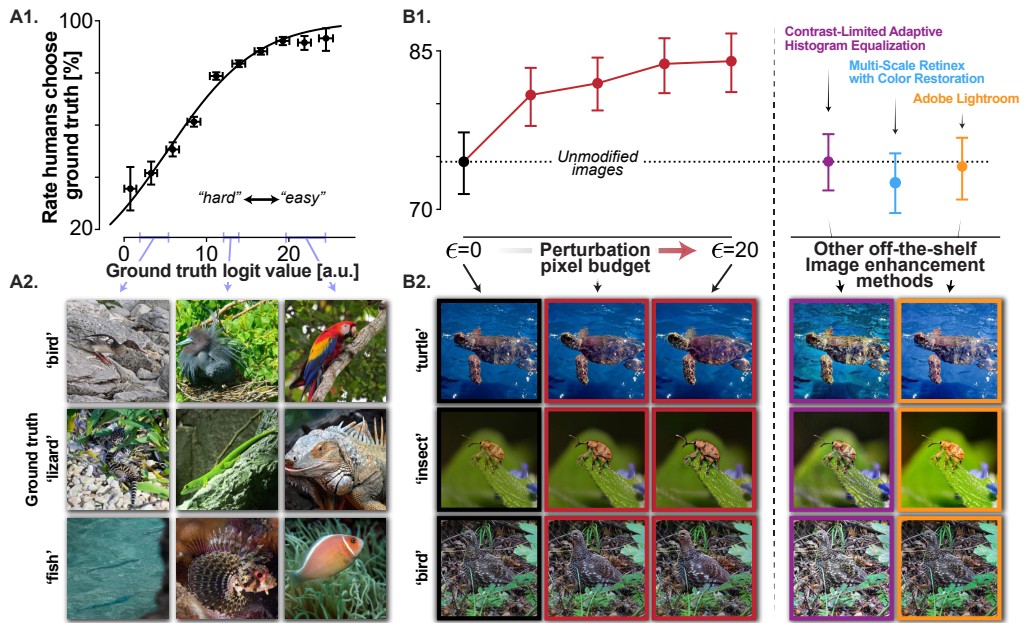

Figure 1: **Robustified ANNs can be used out-of-the-box as image recognition difficulty estimators and ground truth percept enhancers.** *We consider a 16-way basic animal classification task. Panel A1 shows the correspondence between human categorization accuracy and model-computed ground truth logit activation values. The curve denotes a logistic regression model predicting the probability of a correct response using only the logit value ($p < 0.001$ from the Wald statistic, $AUC = 0.72$ under 10-fold cross validation). A2 shows example images with varying ground truth logit values (predicted difficulty). B1 shows how perturbing images via ground truth logit maximization increases human recognition accuracy progressively with the $\ell_2$-norm perturbation pixel budget $\epsilon$. Other off-the-shelf image enhancement methods do not increase categorization accuracy, despite inducing larger perturbations of $\epsilon = 43$, $\epsilon = 106$, and $\epsilon = 26$ on average from left to right. B2 shows example images: unmodified (left), enhanced by ground truth logit maximization with pixel budgets $\epsilon = 10$ and $\epsilon = 20$ (middle), and enhanced by baseline off-the-shelf methods (to the right of the dotted line). All vertical error bars are 95% confidence intervals by bootstrap. Horizontal error bars in panel A1 show the standard deviation among images within each logit value bin.*

We identify a potential strategy to accelerate the visual learning process by extrapolating from the perceptual learning literature. Simple visual tasks, such as line orientation discrimination, reveal a curriculum effect whereby providing easy examples to a novice human learner, before gradually increasing the difficulty, promotes faster perceptual learning (Lu & Dosher, 2022). Motivated by these findings, we ask whether the human-aligned nature of robustified ANNs allows them to augment human learning in complex image domains, chiefly by reducing the initial task difficulty and then increasing the difficulty as learning progresses. A demonstration that these models can be used to enhance learning serves as an additional scientific test of the models' alignment with human perception, while also pioneering a potentially beneficial application of ANNs in education.

To test the viability of enhancing image category learning in humans, we first establish two key empirical observations, summarized in Fig. 1: (i) we find that the human error rate in an image categorization task is strongly predicted by the ground truth logit activation value of a robustified ANN, making it a valid image recognition difficulty score for humans; (ii) we find that this relationship also holds in reverse – pixel-level perturbations can be generated by the model to maximize the ground truth logit activation, producing an "enhanced" version of the image that is easier for humans to recognize as the ground truth category. While previous findings demonstrate that model-guided perturbations can modulate human perception *away* from the ground truth category (Gaziv et al., 2023), we conversely seek to *amplify* category-relevant features to facilitate correct classification. We observe that our model-based image enhancements yield significant increases in human accuracy on a classification task with basic animal categories, unlike conventional image enhancement techniques that adjust low-level image properties such as lighting and contrast. We thus propose an algorithm that combines model-based image enhancement and image difficulty prediction to generate optimized curricula for novice humans learning challenging image categorization tasks.

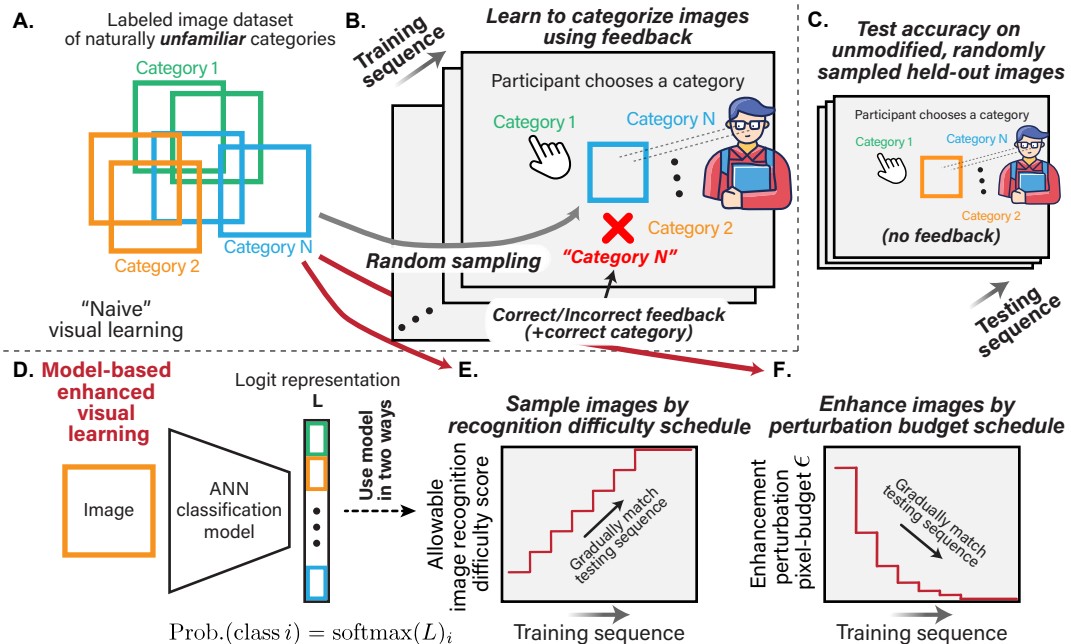

Figure 2: **Robustified ANNs can be used to boost image category learning in humans.** *A novice human learner undertakes a challenging image categorization task, which consists of a training phase (**B**) and a test phase (**C**). Images for both phases are randomly drawn from a labeled image dataset of unfamiliar fine-grained categories (**A**). Feedback (correct/incorrect, with indication of the correct category) is delivered after each trial during the training phase only. Our proposed "Logit-Weighted Image Selection and Enhancement" (L-WISE) approach uses an ANN model (**D**) to augment the visual curriculum by using the difficulty score to sample images based on a predefined increasing schedule of maximal difficulty per trial (**E**), and by enhancing images for easier recognition with an enhancement magnitude that decreases along a predefined schedule (**F**).*

Our proposed method, "Logit-Weighted Image Selection and Enhancement" (L-WISE), is illustrated in Fig. 2. In our primary experimental setting, a human participant learns an unfamiliar image categorization task by viewing a series of examples, providing a category judgment for each image before receiving feedback indicating the correct category (Fig. 2A,B). Upon completion of the training phase, test accuracy is measured on held-out images in similar trials without feedback (Fig. 2C). L-WISE intervenes on this naive visual learning baseline by using a robustified ANN model in two ways: (i) it uses the ground truth logit difficulty score to sample training images based on a predefined, *increasing* schedule of maximal difficulty per trial (Fig. 2D,E); (ii) it enhances training images with a perturbation magnitude that *decreases* along a similarly predefined schedule (Fig. 2D,F).

Despite the human visual system being well-adapted for rapidly learning new visual categories, we find that L-WISE gives rise to substantial accuracy gains of 33-72% in test-time accuracy margins above chance relative to control participants. In addition to improved accuracy, L-WISE also significantly reduces the training phase duration by 20-23% with a constant number of trials. We demonstrate these effects across three varied image domains and category spaces: moth species in natural photographs, skin lesions in dermoscopy images, and pathologic findings in histology images.

Our main contributions are:

• We establish a new state-of-the-art in predicting image recognition difficulty for humans, using the robustified ANN ground truth logit activation value as a simple but effective difficulty metric.

• We show that robustified ANNs can guide image perturbations that enhance the ability of humans to accurately report the associated ground truth category label.

• We propose a novel model-based visual learning augmentation approach for humans that substantially increases test-time categorization accuracy and also reduces training time. To the best of our knowledge, this is the first application of image enhancement to augment human visual learning.

• We demonstrate the broad applicability of our proposed method in a variety of image domains, including clinically relevant dermoscopy and histology categorization tasks.

## 2 RELATED WORK

We develop two important capabilities that form the foundation of our approach to assisting human learners: (1) state-of-the-art predictions of the recognition difficulty of images for humans, and (2) image perturbations that increase human categorization accuracy. Many works have ranked the difficulty of images to design curricula for training ANNs (Wang et al., 2021b). Leading approaches include the c-score learning speed proxy (Jiang et al., 2021) and the prediction depth (Baldock et al., 2021) calculated for each image. Mayo et al. (2023) applied both of these techniques to predict the recognition difficulty of natural images for humans, defined as either the minimum viewing time required to classify a given image correctly, or (as in our work) the proportion of humans who correctly classify it. Here, we show that a robustified ANN model's logit score associated with the ground truth class is a more accurate predictor of image difficulty for humans than prior methods.

Enhancing image quality has been the focus of many previous studies (Qi et al., 2021), ranging from correction of low-level properties such as lighting and contrast (e.g., Zuiderveld (1994); Jobson et al. (1997)) to ANN models that "upsample" images to higher resolutions (Anwar et al., 2020). However, very little research has focused on enhancing images to more strongly represent a specific category. Previous works in this vein focused on making images easier for ANN models to correctly classify (Kim et al., 2023; Tussupov et al., 2023) or less vulnerable to subsequent adversarial attacks (Salman et al., 2021; Frosio & Kautz, 2023). However, such methods do not strongly affect human responses due to perceptual misalignment between humans and naive ANNs (Gaziv et al., 2023).

Many studies have focused on model-human alignment. Brain-Score benchmarks models in terms of neural representations and behavior (Schrimpf et al., 2018). "Harmonization" methods drive alignment using an auxiliary objective on ANN-predicted feature importance maps and crowd-sourced human maps (Fel et al., 2022). Other works introduce architecture components to account for additional aspects of human vision, such as the dorsal-stream pathway (Choi et al., 2023), or recurrent connections for contextual reasoning (Bomatter et al., 2021) and visual search (Gupta et al., 2021).

A key property that enables ANNs to generate human-interpretable image perturbations is that of perceptually aligned gradients, which is closely related to adversarial robustness and can be induced through adversarial training (Ganz et al., 2023; Gaziv et al., 2023). In our present work, we apply adversarially-trained ANNs to enhance images such that they are more strongly associated with their ground truth label by the guiding model and by humans. To the best of our knowledge, we are the first to demonstrate improved human performance on image classification tasks through category-specific image enhancement.

Our primary goal is to apply difficulty prediction and image enhancement to augment human learning. The emerging field of machine teaching (Zhu, 2015) employs machine learning to find or generate optimal "teaching sets" that can be used to train models or humans. While many such approaches have been successfully applied to training machine learning models (e.g., Liu et al. (2017); Qiu et al. (2023)), few studies have successfully enhanced image category learning in humans and most of these focus on teaching set selection. Singla et al. (2014) propose STRICT, which optimizes the expected decrease in learner error based on how the selected images and their labels constrain a linear hypothesis class in a feature space. Johns et al. (2015) extend a similar approach to select images in an online fashion by modeling the learner's progress. MaxGrad (Wang et al., 2021a) uses bi-level optimization to iteratively refine a teaching set by modeling learners as optimal empirical risk minimizers. Most similar to our work are approaches like EXPLAIN (Mac Aodha et al., 2018), which uses ANN class activation maps (CAMs) to highlight relevant image regions while providing feedback to the learner. EXPLAIN also selects a curriculum of images based on (i) a multi-class adaptation of STRICT, (ii) representativeness (mean feature-space distance to other images of the same class), and (iii) the estimated difficulty (entropy) of the CAM explanations. Chang et al. (2023) use bounding boxes to highlight image regions attended to by experts and not novices, allowing humans to more accurately match bird or flower images to one species among five shown in a gallery.

Our approach to learning augmentation departs from previous studies in several ways. We make explicit estimates of image difficulty with unprecedented accuracy to select easier images for early-stage learners. Our work is unique in employing category-specific image enhancement, which is a novel technique in itself, to improve the teaching efficacy of a given set of images. While Mac Aodha et al. (2018) and Chang et al. (2023) help learners by explicitly highlighting *where* learners should attend to in each image, we take a distinct and complementary approach by implicitly highlighting *what* learners should attend to in order to classify images correctly.

## 3 OVERVIEW OF APPROACH AND EXPERIMENTS

Our approach to improving visual learning in humans is based on two key observations regarding human-aligned ANN models of the ventral visual stream: (i) they can accurately predict the recognition difficulty of specific images for humans (Fig. 1A), and (ii) they can be used to perturb images in a way that enhances the ability of humans to accurately report the original ground truth category (Fig. 1B). In other words, these models can be used out-of-the-box as *category-recognition difficulty estimators* and *category-percept enhancers*. As such, we propose using them to design sequences of images that humans can use to more efficiently learn to recognize unfamiliar image categories. We test our approach in a variety of challenging image domains: natural photographs (moth species classification), dermoscopy images (skin lesion classification), and histology images (benign vs. pre-cancerous colon tissue classification).

### 3.1 TRAINING TASK-SPECIFIC ROBUSTIFIED MODELS

To obtain robustified models for task-specific category spaces, we adversarially trained ResNet-50 ANNs (He et al., 2016) on the ImageNet-1K (Deng et al., 2009) and iNaturalist 2021 (Van Horn et al., 2021) datasets (separately) using the same techniques as Mądry et al. (2018). To adapt the resulting model to the three categorization tasks of interest, we conducted additional adversarial fine-tuning on the corresponding smaller datasets: a subset of moth species images from iNaturalist (after iNaturalist pretraining), the HAM10000 dermoscopy dataset (Tschandl et al., 2018) (ImageNet pretraining), and the MHIST histology dataset (Wei et al., 2021) (ImageNet pretraining).

### 3.2 PREDICTING CATEGORY RECOGNITION DIFFICULTY

We propose a simple approach to predicting the human categorization error rate on specific images, in the form of a new image recognition difficulty score: the logit activation (pre-softmax) at the ground truth category output unit of a robustified ANN. The higher this logit value is, the lower the human categorization error rate. We establish this relationship through human participant responses during a natural image categorization task with 16 basic animal categories (Fig. 1A). We found this logit score to be the new state-of-the-art in predicting human error rates (see Appendix Fig. S6).

### 3.3 GENERATING IMAGE PERTURBATIONS TO ENHANCE CATEGORY PERCEPTS

While the ground truth logit score predicts image difficulty at baseline, we show that we can also generate perturbations that maximize this value by backpropagating from the logit score to pixel space and running projected gradient ascent to update the pixel values, limiting the "pixel budget" $\ell_2$ norm of the perturbation to a value $\epsilon$. These perturbations make images easier for humans to recognize with respect to their ground truth labels (Fig. 1B). This approach is analogous to Gaziv et al. (2023), but is designed to enhance the ground truth percept rather than guiding away from it.

### 3.4 BOOSTING IMAGE CATEGORY LEARNING IN HUMANS

Our approach to augmenting visual learning is summarized in Fig. 2. In the naive baseline scenario (Fig. 2A-C), the novice human learner is presented with a sequence of training trials through an online platform. In each training trial, a randomly selected image from one of $N$ categories is presented, and the learner attempts to choose the correct category among $N$ possible labels. The learner receives feedback after each training trial indicating the correct category label and whether or not their response was correct. In most of our experiments, to ensure that participants begin at the chance level and to avoid possible priors induced by the category names, we assign each category to a random Greek name unrelated to the task (with random assignments for each participant). After completion of the training phase, the experiment transitions into a test phase, in which the same task continues over a held-out set of images and no feedback is provided (Fig. 2C). During the test phase we measure the main visual learning outcome of interest, the test accuracy.

Harnessing key observations of robustified ANNs, our "L-WISE" approach uses one such model to optimize the training phase in a way that improves the test accuracy via two mechanisms: (i) sampling training images based on their predicted recognition difficulty, and (ii) enhancing the training images. We adjust the "strength" of both mechanisms in a time-dependent manner: for the former, we set the maximum allowable difficulty score for an image to be shown in a given trial, and for the latter, we limit the enhancement perturbation magnitude to an $\ell_2$ norm pixel-budget $\epsilon$. The user of our approach can flexibly define arbitrary time-dependent profiles for image selection and enhancement (Fig. 2D-F). In this study, we used a step-wise linear ramp schedule for the allowable image difficulty at a given time, and exponential tapering of the enhancement $\epsilon$. Intuitively, this should

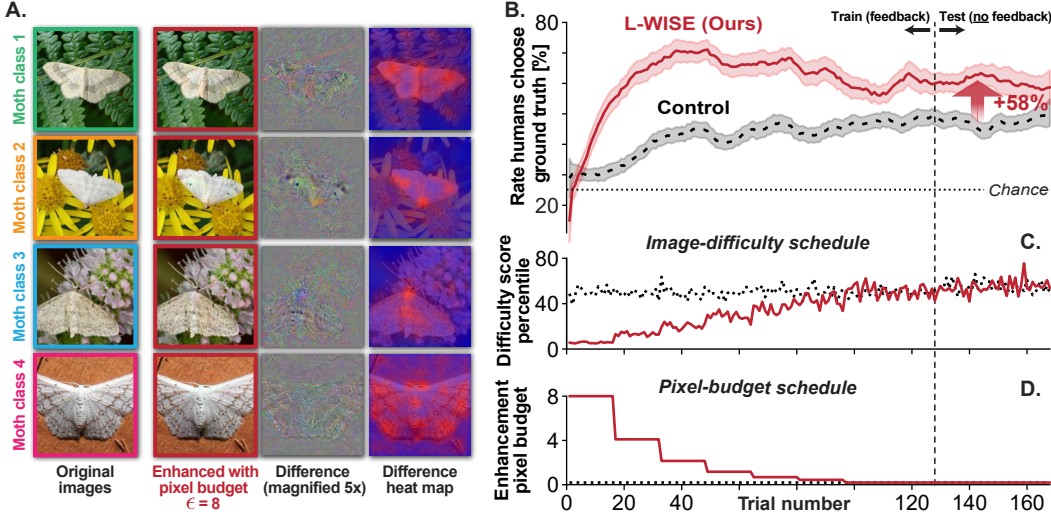

Figure 3: **Novice learners who had their curriculum augmented by our method showed improved test-time categorization accuracy for previously unfamiliar categories.** *This figure shows empirical results from a 4-way fine-grained moth species classification task. Panel **A** shows examples of the 4 moth classes, side-by-side with their model-enhanced versions at the highest pixel budget used in our experiments ($\epsilon = 8$). While subtle, one notable difference is the distinctive wing spots of moth class 2, which are enlarged in the enhanced version of the image. Also included are difference images showing the (5x magnified) difference between original and enhanced images, and heat maps with more red coloration in regions of larger changes from enhancement. **B** compares the average smoothed accuracy of participants in the L-WISE group and a control group. Shaded areas denote the standard error of the mean. The test accuracy gain of the L-WISE group relative to the control group is statistically significant ($\chi^2(1)$ test, $p < 0.001$). **C, D** show the trial-dependent empirical profiles of the average image difficulty percentile of selected images, which (noisily) increases step-wise, and the perturbation pixel budget for enhancement ($\epsilon$), which decreases step-wise. These profiles are uniform in the control group (black dotted lines), denoting randomly-chosen non-enhanced images.*

correspond to an easy-to-challenging trend during the training phase. Extensively optimizing these schedules was not a goal of our study, which serves primarily as a proof-of-concept. Applying image selection and enhancement led to significant gains in the accuracy of human participants on unmodified, randomly-selected test images, while also reducing the time needed to complete the training phase (which consists of a constant number of trials). This result was robustly observable across the varied image domains and category spaces we tested.

# 4 RESULTS

We used robustified ANNs to both enhance images and predict the difficulty of images across multiple domains. We applied both of these techniques to improve the final test performance (on unmodified, randomly-chosen images) of novice humans learning challenging image classification tasks.

## 4.1 ROBUST MODELS CAN BOTH PREDICT IMAGE RECOGNITION DIFFICULTY AND REDUCE IT

We tested the effects of a novel model-based image enhancement algorithm on human image categorization accuracy. We demonstrate that we can enhance images by maximizing the ground truth logit from a robustified ANN (ResNet-50) using gradient ascent in image pixel space. As the magnitude of the enhancement perturbations grows ($\ell_2$ norm pixel budget $\epsilon$), human participants become increasingly accurate on a 16-way animal photograph classification task derived from ImageNet (Fig. 1B, chance = 1/16). While mean accuracy on the original, unmodified ($\epsilon = 0$) images was 0.75, mean accuracy on enhanced images was as high as 0.84 (at $\epsilon = 20$). The accuracy gains from enhancement appear to approach a saturation point as the perturbations grow larger. The improvements in accuracy are also somewhat dependent on the starting ground truth logit score, as shown in Appendix Fig. S11: accuracy gains are larger for "difficult" images than for "easy" images. Baseline enhancement algorithms Contrast-Limited Adaptive Histogram Equalization (CLAHE, Zuiderveld (1994)), Multi-Scale Retinex with Color Restoration (MSRCR, Jobson et al. (1997); Petro et al.

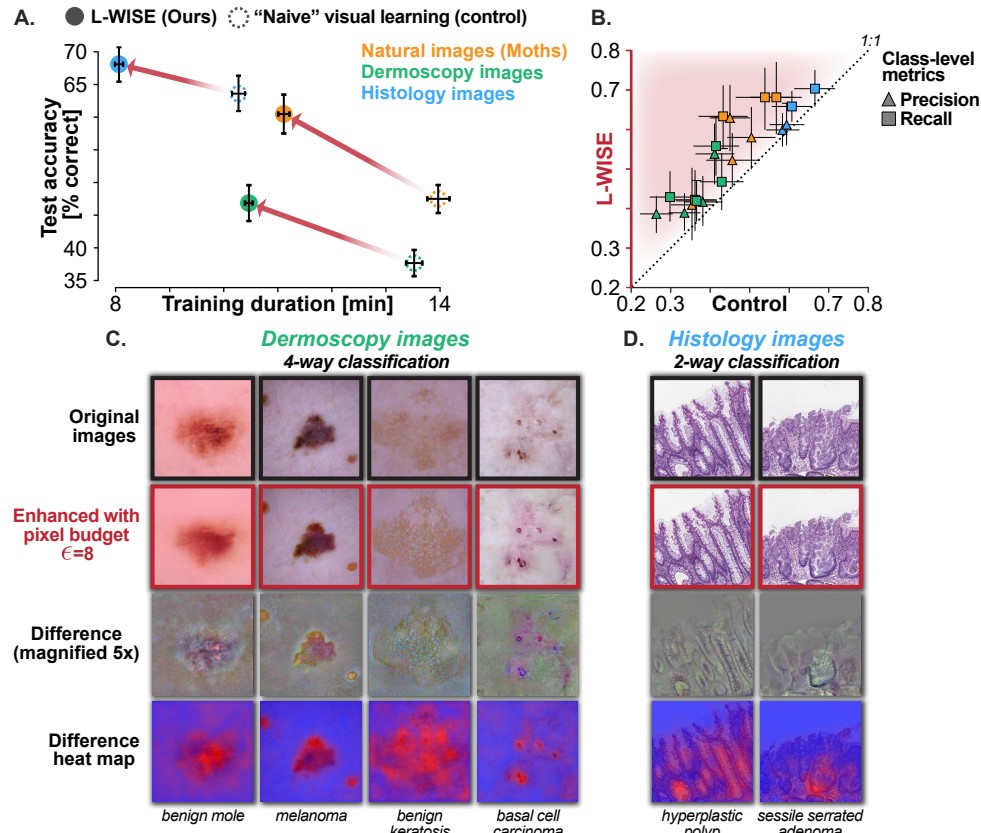

Figure 4: **Our approach can boost time efficiency and final accuracy of image category learning for humans across varied image domains, including in clinically relevant tasks.** *Panel A compares the mean test-phase accuracy and training-phase duration of human participants who were randomized to L-WISE or control groups and learned a moth photo, dermoscopy, or histology classification task. All differences between L-WISE and the control group are statistically significant ($\chi^2(1)$ test, $p < 0.05$). Panel **B** shows precision and recall in L-WISE and control groups, with each point representing a specific class in one of the three tasks. All error bars show 95% bootstrap confidence intervals. Each class from the dermoscopy and histology tasks is illustrated in panels **C** and **D** respectively, similarly to the moth classes in Fig. 3A.*

(2014)), and Adobe Photoshop Lightroom's "Auto" enhancement feature (LR, Adobe Inc. (2024)) had no significant effect on human accuracy, despite inducing image perturbations of considerably larger $\ell_2$ norm on average than the $\ell_2$ pixel budget $\epsilon$ values we used for model-based enhancement.

We also demonstrate that the robustified model's ground truth logit $L_{\mathrm{gt}}(x)$ is strongly correlated with the rate at which humans choose the ground truth category associated with image $x$ in a 16-way basic animal classification task (Fig. 1A). We used robustified ResNet-50 to calculate $L_{\mathrm{gt}}$ for each of the 2,400 distinct natural images used in the task, and applied logistic regression to predict binary correct vs. incorrect responses to individual image trials. We pooled responses to original images with those to modified control-group images that were not enhanced by robust models, recomputing $L_{\mathrm{gt}}$ for each image version (see also Appendix Fig. S5). The logistic regression model (Fig. 1A) used $L_{\mathrm{gt}}$ to predict the binary correctness of the trial responses with Area Under the Receiver Operating Characteristic Curve (AUC) = 0.72 (10-fold cross-validation, $p < 0.001$ via Wald statistic). Notably, we find that this simple approach predicts the difficulty of individual images for humans more accurately than existing state-of-the-art metrics (Mayo et al. (2023), see Appendix Fig. S6). In addition to ResNet-50, we demonstrate both difficulty prediction and image enhancement with XCiT vision transformers (Ali et al. (2021), Appendix Figs. S7-S10).

## 4.2 L-WISE IMPROVES BOTH TEST ACCURACY AND LEARNING SPEED FOR HUMANS

We applied both image difficulty prediction and image enhancement as part of a novel method that designs image sequence curricula for novices learning challenging image classification tasks.

|  | *Idaea* moth photos | | Skin lesion dermoscopy images | |
|---|---|---|---|---|
|  | Mean acc. | Training duration | Mean acc. | Training duration |
| Chance level | 0.25 | - | 0.25 | - |
| Control | 0.47 (0.45, 0.50) | 14.0 (13.8, 14.2) | 0.38 (0.36, 0.40) | 13.5 (13.4, 13.7) |
| ET | 0.58* (0.55, 0.61) | 11.8 (11.7, 11.9) | 0.45* (0.43, 0.47) | 13.1 (12.9, 13.3) |
| ET (shuffled) | 0.53* (0.50, 0.56) | 15.1 (14.8, 15.4) | 0.39 (0.36, 0.42) | 13.3 (13.2, 13.4) |
| DS | 0.49 (0.47, 0.52) | 13.9 (13.7, 14.1) | 0.44* (0.42, 0.48) | 11.5 (11.4, 11.6) |
| DS (shuffled) | 0.58* (0.55, 0.60) | 12.6 (12.5, 12.8) | 0.45* (0.42, 0.48) | 11.0 (10.9, 11.1) |
| L-WISE | **0.60* (0.58, 0.64)** | **11.1 (11.0, 11.2)** | **0.47* (0.44, 0.50)** | **10.5 (10.4, 10.5)** |

Table 1: **Both image enhancement tapering (ET) and image difficulty selection (DS) contribute to the ability of L-WISE to assist learners.** The benefits of image enhancement are dependent on easy-to-hard sequencing ("ET" outperforms "ET (shuffled)"), but the benefits of difficulty-based selection appear to stem from simply showing an easier distribution of images during training ("DS (shuffled)" performs as well as or better than "DS"). Training durations are in minutes. Values in parentheses show 95% confidence intervals from 10,000 bootstrap replicates. * denotes significant differences in accuracy from the control group ($p < 0.01$, $\chi^2(1)$ test).

Our proposed algorithm, "Logit-Weighted Image Selection and Enhancement" (L-WISE), operates on image trial sequences used to train human participants on image classification tasks through trial-by-trial feedback. The performance of each participant is evaluated in a subsequent testing phase without feedback, which includes only randomly-selected, unmodified images (unaffected by L-WISE). During the early portion of the training phase, L-WISE randomly selects images from below a certain difficulty percentile that stepwise-linearly increases as the training phase progresses. Selected images are enhanced at each trial during this period, via perturbations within an $\ell_2$ pixel budget $\epsilon$ that decreases in a stepwise-exponential manner (Figs. 2E-F and 3C-D).

We tested L-WISE's efficacy at improving test-time accuracy and training duration of human learners on three challenging image category learning tasks (Figs. 3-4). Participants were randomly assigned to a control group with randomly-selected, non-enhanced images throughout the task, or to an L-WISE group. L-WISE increased the average test-time accuracy margin above chance levels by 57.6% on a 4-way moth species classification task ($p < 0.001$ on $\chi^2(1)$ test), by 72.3% on a 4-way skin lesion dermoscopy task ($p < 0.001$), and by 33.1% on a binary colon histology task ($p = 0.023$) (Fig. 4A). In all three tasks, participant accuracy in the L-WISE group increased initially and then declined to varying degrees as more and more difficult images were selected and the degree of enhancement was simultaneously reduced. In addition to improving test-time accuracy, L-WISE decreased the mean time to learn the task (with a fixed number of training trials) by 20% for the moth task, 23% for the dermoscopy task, and 22% for the histology task (Fig. 4A).

### 4.3 IMAGE ENHANCEMENT AND SELECTION BOTH CONTRIBUTE TO EFFICACY OF L-WISE

We tested several ablated versions of L-WISE (in the moth and dermoscopy tasks) to determine the relative contributions of its components: (A) image enhancement based on logit maximization, (B) selection of images according to logit-estimated difficulty, and (C) easy-to-hard curriculum trends enabled by A and B (Table 1). In "Enhancement Tapering" (ET), only the enhancement component of L-WISE is active, with random image selection as in the control group. Conversely, in "Difficulty Selection" (DS), images are selected based on difficulty but not enhanced. In "ET (shuffled)" and "DS (shuffled)," after applying ET or DS, the ordering of affected training trials is randomly permuted. Shuffling flattens easy-to-hard trends, isolating effects from (i) the mere presence of enhanced images in ET, and (ii) seeing easier images on average in DS (DS limits max. difficulty of early images, so DS/DS (shuffled) have easier training images than Control on average; see Fig. 3C).

The results show that both ET and DS have significant benefits in isolation. ET increased the test-phase accuracy margin above chance by 46.8% for the moth task and 56.5% for the dermoscopy task, while DS increased the same margin by 8.1% (not significant) and 53.2% respectively. ET (shuffled) was less effective, increasing the margin above chance by 23.0% for the moth task and 11.2% (not significant) for the dermoscopy task. Surprisingly, DS (shuffled) outperformed DS without shuffling, increasing the margin above chance by 45.2% for the moth task and 58.2% for the dermoscopy task (the increase of DS (shuffled) relative to DS is statistically significant for the moth task only). Unablated L-WISE numerically outperformed all ablated conditions, increasing the margin above chance

by 57.6% for the moth task and 72.3% in the dermoscopy task. However, additional paired comparisons indicated that the differences between these increases and those from ET or DS (shuffled) are not statistically significant for either task. ET and DS did demonstrate a statistically significant additive benefit in terms of learning speed, however. On the moth task, training duration was 6% shorter for L-WISE than for the next fastest group, which was ET. Similarly, on the dermoscopy task, training duration was 5% shorter for L-WISE than for the next fastest group, DS (shuffled).

## 5 DISCUSSION

In this study, we demonstrate that robustified ANNs can be used to both predict the empirical recognition difficulty of individual images for humans, and also generate enhanced versions of images that are easier for humans to correctly categorize. We harness these capabilities to develop a model-based curriculum design algorithm to augment human image category learning. We show that a combination of selecting images within a certain difficulty range, and perturbing those images to enhance the perception of the ground truth category, leads to substantial improvements in human training speed and test-time classification accuracy on randomly-selected, unmodified images.

The results of our ablation study show that at least a portion of these improvements can be achieved with image enhancement alone: humans can learn from perturbed images and subsequently achieve superior generalization to unseen examples (Table 1). There are several possible explanations for this effect. Image enhancements might draw the learner's attention to relevant features, such as the distinctive dot in the middle of each wing of the *Idaea biselata* moth in Fig. 3A (Hufnagel, 1767), or the irregular border and multiple colors that appear to be enhanced in the melanoma image in Fig. 4C (Tsao et al., 2015). Enhancements might also diminish features that distract from or contradict the ground truth: for example, in Appendix Fig. S11B (second image from the left), buffalo standing behind the ground truth "antelope" are variously blurred or nearly erased. Analogously, images with high ground truth logits (low predicted difficulty) might tend to have clearer class-relevant features and fewer distracting or contradictory features. These parallel explanations for the effects of image enhancement and image selection could help clarify why the "enhancement tapering" and "difficulty selection" strategies in isolation provide comparable accuracy gains to each other, and why combining both strategies did not lead to large additive improvements in accuracy (although additive increases in learning speed were observed).

### 5.1 LIMITATIONS

This work is a proof-of-concept demonstration that robustified ANNs can be applied to augment image category learning in humans. We did not exhaustively search for optimal curriculum design strategies or image enhancement hyperparameters, nor did we study the "dose-dependency" of image enhancement or selection. L-WISE applies a fixed schedule of maximal image difficulty and image enhancement magnitude for all learners: human learning could plausibly be augmented more effectively by adapting the degree of image enhancement and the difficulty of selected images to the learner's progress in real time (Lu & Dosher, 2022; Mettler & Kellman, 2014).

One caveat to our approach is that logit maximization can sometimes appear to have a homogenizing effect on image distributions. For example, "benign mole" dermoscopy images enhanced with high $\epsilon$ budgets tend to all resemble smooth and uniform blobs (see Appendix Fig. S12H for an example). This clearly illustrates a task-relevant difference from melanoma (which tends to be asymmetric with irregular borders (Tsao et al., 2015)), but obscures much of the real-world heterogeneity among benign moles. A similar risk might apply to image selection: images with high ground truth logits might belong to limited regions of the overall class distribution. Biased perturbations or selections reflecting biases in the underlying datasets are another concerning possibility, particularly for dermoscopy (Daneshjou et al., 2022). These caveats must be thoroughly investigated before deployment of real-world educational applications of L-WISE, especially for clinical tasks with potential patient safety ramifications.

## 6 METHODS

***Predicting Image Difficulty.*** To predict the relative difficulty $d \in [0, 1]$ of each image, we extract the logit value corresponding to the image's ground truth class ($L_{\text{gt}}$) immediately upstream of the final softmax function. We sort the logits in descending order such that $L_{(1)} \geq L_{(2)} \geq ... \geq L_{(n_{c,s})}$. $n_{c,s}$ is the number of images for a given class $c$ and training/validation/testing split designation $s$. We calculate class-specific difficulty percentile $d_j$ for image $j$ using the equation $d_j = \text{rank}(L_{(j)})/n_{c,s}$.

***Generating Perturbations to Enhance Images.*** To enhance an image using a pretrained ANN, we maximize $L_{gt}$ through projected gradient ascent, onto a hypersphere of radius $\epsilon$, in pixel-space (see Appendix Section S1 for model training and projection details). In some cases, we also explicitly minimize the logit of competing classes. We generate perturbed image $x'$ via the optimization:

$$x' = x + \arg\max_{\|\delta\| < \epsilon} \left( L_{gt}(x + \delta) - \frac{\alpha}{|C| - 1} \sum_{c \in C: c \neq gt} L_c(x + \delta) \right) \tag{1}$$

In Equation 1, $\delta$ is a perturbation tensor of the same dimensionality as $x$ and with an $\ell_2$ norm less than pixel budget $\epsilon$. $L_{gt}$ is the ANN's logit score associated with the ground truth class, and $L_c$ is the logit associated with class $c$ ($C$ is the set of all classes, with cardinality $|C|$). $\alpha$ determines the extent to which logits for competing classes are minimized. We set $\alpha = 0$ for the ImageNet animal classification experiments and $\alpha = 1$ for the three fine-grained image category learning experiments.

***Image classification and learning experiments with human participants.*** We recruited 521 human subjects using the online platform Prolific. We allowed subjects to participate in multiple experiments, but only once for each of the three learning tasks. We used the jsPsych library (De Leeuw, 2015) with the jspsych-psychophysics plugin (Kuroki, 2021) for all experiments. All experiments included 10-12.5% attention check trials where the subject classifies an image of a circle or triangle (e.g., see Appendix Fig. S4). We analyzed data from subjects with $\geq 90\%$ attention check accuracy.

To measure the effects of enhancement on a presumably already-learned task, we tested the accuracy of human subjects at classifying 16 basic types of animals (frog, bird, dog, etc., see Appendix Section S3). Images were shown for 17 milliseconds each, after which the participant was given 15 seconds to respond. All images were drawn from the validation set of ImageNet (Deng et al., 2009). Appendix Fig. S3 shows screenshots of the task as it appeared to participants. In our main experiment with this task (Fig. 1), each subject viewed interspersed images from 9 conditions: original images, images enhanced by maximizing $L_{gt}$ with $\epsilon = 5, 10, 15$, and 20, images enhanced with one of three off-the-shelf baseline methods, and images disrupted by minimizing $L_{gt}$ (internal control). Participants were notified after each incorrect response and given a small monetary bonus for each correct response (except for disrupted images). 62 participants viewed 18 images for each of 16 classes, 2 from each condition, in a shuffled ordering for a total of 288 trials per participant and 17,856 overall. These main trials followed a screening phase with 32 trials (200ms presentation times, $\geq 24$ correct with $\geq 1$ correct per class required to proceed, multiple screening attempts allowed) and a 32-trial warm-up phase with 17ms image presentations. Screening and warm-up phases used original, unmodified images, and data from these phases were not included in any analyses.

The image category learning tasks consisted of either 4 (moths, dermoscopy) or 2 (histology) image classes. Participants were shown each image for up to 10 seconds and used the mouse (4-way tasks) or keyboard (binary task, "F" and "J" keys) to respond. After each trial, the participant was notified of the ground truth label and whether their response was correct (screenshots in Appendix Fig. S4). Each session consisted of 8 training blocks of 16 trials each (4 per class, or 8 per class for histology), and 2 testing blocks of 20 trials each during which no post-trial feedback was provided. Each block contained an equal number of images from each class in a random order. Participants were informed upon recruitment that they could receive a progressively higher monetary bonus if their test-phase accuracy exceeded certain thresholds. Before participating in the main learning tasks, subjects had to first learn an easier binary classification task (leatherback vs. loggerhead turtles) and respond correctly to at least 7 of 8 test-phase trials. We randomly assigned ~30 participants per experimental condition, except the ablation study control groups of ~60 participants (overall min. 27, max. 68).

***Assisting learners with the L-WISE algorithm.*** L-WISE consists of two strategies applied in parallel: Enhancement Tapering (ET) and Difficulty Selection (DS). Both strategies operate only on images in the first 6 of 8 trial blocks in the training phase. In ET, we enhanced images in the first block of training-phase trials with $\epsilon = 8$. $\epsilon$ is halved for each subsequent block until it is set to 0 after the 6th block. In DS, only images with $d < d_b$ were sampled for each block $b$. $d_b$ was incremented by 0.15 at the end of each of the first six blocks, beginning at $d_1 = 0.1$ and reaching $d_7, d_8 = 1.0$. Determining an effective schedule of $\epsilon$ and $d_b$ did not require extensive hyperparameter tuning. After a pilot experiment in which we decreased $\epsilon$ linearly starting from $\epsilon = 20$, (see Appendix Fig. S12H-M), we switched to the $\epsilon$ schedule above and changed no other hyperparameters for any of the three learning tasks/image domains. In the "shuffled" versions of DS and ET (Section 4.3), images in blocks 1-6 are selected or enhanced before a constrained shuffling procedure, whereby each image in blocks 1-6 may switch positions with any other of the same class regardless of $d$ or $\epsilon$.

## 7 ETHICS STATEMENT

This study involved experiments with human participants conducted over the internet, using the Prolific platform for the main experiments and Amazon Mechanical Turk for pilot experiments. We followed a study protocol approved by Boston Children's Hospital's Institutional Review Board. Participants provided informed consent before participating in any experiments. The experiments posed no greater than minimal risk to the participants. All participants were anonymous, and all data is de-identified. Participants were provided with our contact information, and that of the Office of Clinical Investigations at Boston Children's Hospital, for any questions or concerns about the study. We calibrated the participant compensation amounts for each experiment to meet or exceed the equivalent of $15.00 USD per hour, including during screening tasks. Participants were recruited using the "Standard Sample" option in Prolific, and were diverse in gender, age, and race/ethnicity (please see Table S2 for a demographic breakdown).

We hope that our work will eventually lead to practical and beneficial applications in education - for example, in the training of doctors in specialties such as pathology, radiology and dermatology where visual perceptual learning is particularly important. We wish to emphasize, however, that more work is needed before our methods can be safely applied in sensitive or high-stakes settings. For example, we apply our approach to improve human performance on a dermoscopy skin lesion classification task derived from the HAM10000 dataset (Tschandl et al., 2018). This dataset is heavily skewed towards images of pale skin, likely a reflection of the lower incidence of skin cancers such as melanoma among people with darker skin tones (Cormier et al., 2006). Models trained on this dataset are known to perform poorly for patients with darker skin (Daneshjou et al., 2022), where melanoma tends to have a different appearance, unfamiliarity with which on the part of clinicians contributes to delayed diagnosis and increased mortality among such patients (Thompson et al., 2023). It is plausible that maximizing the melanoma-associated logit of a robustified model perturbs the images to look more like an average presentation of melanoma (i.e., on light skin), which would risk imparting this bias onto the learner. A similar risk might apply to image selection: images with the highest ground truth logits (which L-WISE presents at the beginning of learning) might tend to belong to specific subclasses or limited regions of the overall class distribution. The possibility of biased perturbations or image selections must be thoroughly investigated in future work before applications of our method in this domain.

## 8 REPRODUCIBILITY STATEMENT

We list all hyperparameters for training robustified neural networks and using them to enhance images in the Appendix. We provide source code that enables reproduction of all data processing steps, experiments, statistical analyses, and figure plots. This includes experiments with human participants: we have developed a flexible framework for automated deployment of web-based image category learning experiments, including hosting tasks as interactive web pages and using cloud-based mechanisms for random group assignment and data collection. The experiments with human participants can therefore be readily reproduced and extended or modified with only a modest amount of configuration required.

ACKNOWLEDGMENTS

This work was supported in part by Harvard Medical School under the Dean's Innovation Award for the Use of Artificial Intelligence, in part by Massachusetts Institute of Technology through the David and Beatrice Yamron Fellowship, in part by the National Institute of General Medical Sciences under Award T32GM144273, in part by the National Institutes of Health under Grant R01EY026025, and in part by the National Science Foundation under Grant CCF-1231216. The content is solely the responsibility of the authors and does not necessarily represent the official views of any of the above organizations. The authors would like to thank Yousif Kashef Al-Ghetaa, Andrei Barbu, Pavlo Bulanchuk, Roy Ganz, Katherine Harvey, Michael J. Lee, Richard N. Mitchell, and Luke Rosedahl for sharing their helpful insights into our work at various times.

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

# Appendix

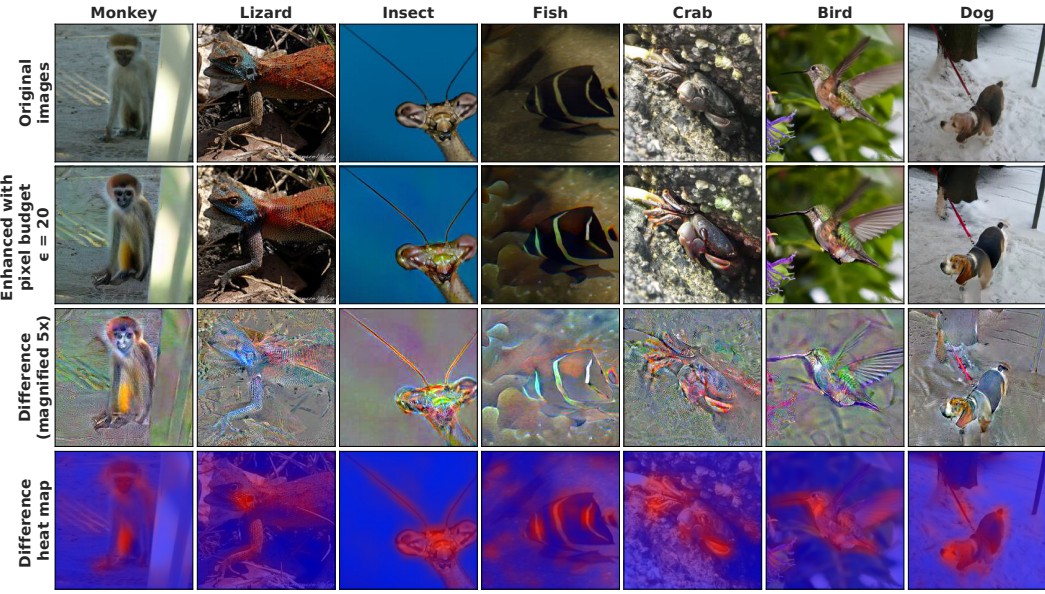

Figure S1: **Ground truth logit enhancement with robustified ANNs leads to semantically meaningful perturbations.** *The top row shows original ImageNet images, and the second row shows the same images after enhancement by robustified ResNet-50 (training $\epsilon = 3$) with a pixel budget of $\epsilon = 20$. The third row shows a 5x magnified version of the difference between the enhanced image and the original, and the bottom row shows a heat map where red regions correspond to larger changes and blue regions correspond to smaller changes.*

## S1 DETAILS ON TRAINING AND USING ROBUSTIFIED GUIDE MODELS

We adversarially trained a ResNet-50 model on ImageNet (Deng et al., 2009), and another on iNaturalist 2021 (Van Horn & Mac Aodha, 2021), with the hyperparameters following Gaziv et al. (2023):

- Epochs = 200
- Base learning rate of 0.1, decreasing by a factor of 10 every 50 epochs
- Batch size = 256
- Weight decay = 0.0001
- Adversarial training $\epsilon = 3.0$ (ImageNet) or $\epsilon = 1.0$ (iNaturalist)
- 7 gradient steps for adversarial attacks
- Adversarial attack step size of 0.5 (ImageNet) or 0.3 (iNaturalist)

The ResNet-50 model adversarially trained on ImageNet was used directly to generate perturbations and difficulty rankings for the 16-way animal classification task, using the logits of the original, fine-grained ImageNet classes (i.e., not the 16 superclasses, "grasshopper" not "insect") for both enhancement and difficulty prediction. The same model was adversarially fine-tuned on the HAM10000 and MHIST datasets before their application (as part of L-WISE) to the dermoscopy and histology tasks respectively. Generally, we trained the models on all available classes in each dataset. For example, we fine-tuned on all 7 classes of the HAM10000 dermoscopy dataset (Tschandl et al., 2018), even though we only used 4 of them in the learning task for humans. When enhancing the images, we include only classes that are part of the experimental tasks as competing classes to have their logits minimized (see main-text Equation 1).

For the moth task, we adversarially fine-tuned the (adversarially) iNaturalist-pretrained model on the four moth classes to be used in the task. We subjectively judged the perturbations from this

fine-tuned model to be of higher quality than those generated using the iNaturalist-pretrained model without fine-tuning, as the four classes of interest are among the 10,000 iNaturalist classes. All adversarial fine-tuning used $\epsilon = 1.0$ with 7 gradient steps of size 0.3. We used learning rates of 0.0001, 0.0004, and 0.001, and batch sizes of 32, 64, and 16, for *Idaea* moth, dermoscopy, and histology fine-tuning respectively. We fine-tuned the entire network end-to-end for each task.

Our choice of $\epsilon = 3$ for ImageNet pretraining follows Gaziv et al. (2023), who found this to be an optimal choice for generating perturbations that disrupt category perception (relative to $\epsilon = 1$ and $\epsilon = 10$). In practice, we found that the models were unable to learn finer-grained tasks with training-time adversarial perturbations as large as $\epsilon = 3$ (iNaturalist pretraining, and fine-tuning on moth photos, dermoscopy images, and histology images) - therefore, we reverted to $\epsilon = 1$ for these settings.

To enhance images in a category-specific manner, we perform the optimization of Equation 1 (main text) in a series of steps using projected gradient ascent (Equation 2), where $k$ denotes the optimization step, $\eta$ the step size, and $\text{Proj}_\epsilon$ a projection onto a hypersphere of radius $\epsilon$ with original image $x$ at its center (see text below Equation 1 for definitions of other symbols).

$$\delta_{k+1} = \text{Proj}_\epsilon\left(\delta_k - \eta\nabla_\delta\left(L_{\text{gt}}(x + \delta_k) - \frac{\alpha}{|C| - 1}\sum_{c\in C: c\neq\text{gt}} L_c(x + \delta_k)\right)\right) \tag{2}$$

Fig. S1 shows several example images enhanced with $\epsilon = 20$ using logit maximization by adversarially-pretrained ResNet-50 ($\epsilon = 3$), along with difference images and heat maps produced by the same method as Figs. 3-4 in the main text. Throughout our experiments, robustified ResNet-50 enhancements with pixel budget $\epsilon$ used ceil($2\epsilon$) steps of $\eta = 0.5$ in a $224 \times 224 \times 3$ pixel space. This formula seems somewhat model-dependent and sometimes requires adjustment: for example, when enhancing images with robustified XCiT (see Figs. S9-S10), ceil($4\epsilon$) steps were required instead of ceil($2\epsilon$) to reach a similar effective perturbation size.

We generate the heat maps in Figs. 3, 4, and S1 by subtracting the enhanced image $x'$ from the original image $x$ element-wise: $\delta = x' - x$. We calculate the magnitude of the changes as $\delta^2$. We apply smoothing using 2D convolution, and normalize the result to have all values between 0 and 1, to produce $\delta_{\text{norm}}$. We then produce the heat map by setting the red channel to $255 \times \delta_{\text{norm}}$ and the blue channel to $255 \times (1 - \delta_{\text{norm}})$. The resulting image shows red in regions where larger changes have taken place, and blue in regions where smaller or no changes have taken place. In Figs. 3, 4, and S1, we superimpose a translucent version of the heat maps ($\alpha = 0.7$) over the original images. Averaging the heat maps across all ImageNet validation set images (Fig. S2) indicates that changes to the images tend to occur in the central regions of the images more than in the periphery, consistent with the observation that our image enhancement approach tends to primarily change image regions corresponding to the main subject of each image.

## S2    DETAILS ON IMAGE PREPARATION FOR EXPERIMENTS

All images presented in all psychophysics experiments were of size $224 \times 224 \times 3$, matching the input dimensions of ResNet-50. Before presentation or any model-based enhancement or difficulty prediction, original images were resized such that the shortest dimension (width or height) was 224 pixels, and then center-cropped to $224 \times 224$. Any single-channel grayscale images were converted to RGB before further processing. The baseline enhancement algorithms Contrast-Limited Adaptive Histogram Equalization (CLAHE (Zuiderveld, 1994)), Multi-Scale Retinex with Color Restoration (MSRCR (Jobson et al., 1997; Petro et al., 2014)), and the "Auto" image tuning feature in Adobe Photoshop Lightroom (Auto-LR (Adobe Inc., 2024)) were applied before the resizing and center-cropping operations.

We generally used images from the validation sets of each dataset for the image category learning experiments, reasoning that the robustified models would be overfitted to training images which could potentially compromise the quality of perturbations and relative difficulty estimates. However, for the moth task, we were limited to 10 validation images per class in the iNaturalist dataset (Van Horn & Mac Aodha, 2021). In this case we used training set images during the training period of the human image category learning experiment and validation images during the test phase. We show that image enhancements are still effective for training set images in Fig. S11A.

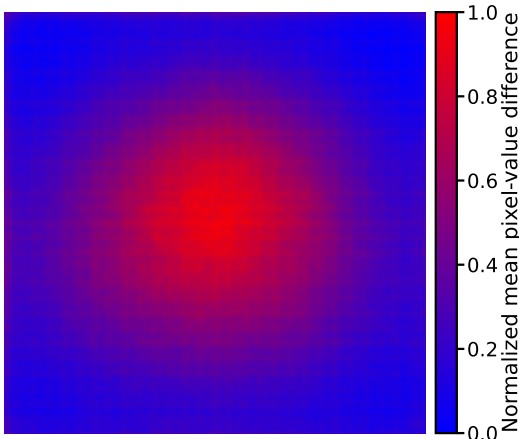

Figure S2: **Pixel-value changes during enhancement of ImageNet images are biased towards the center of the image.** *This heat map indicates which spatial regions of the ImageNet animal images were changed the most on average during logit-maximization enhancement with $\epsilon = 20$. Regions that were changed more on average are more red, and regions that were changed less on average are more blue. The heat map was generated by averaging the normalized absolute pixel value changes across all 2400 ImageNet validation set images that we used for our 16-way animal classification experiments.*

## S3 DETAILS ON 16-WAY IMAGENET ANIMAL CATEGORIZATION EXPERIMENTS

We curated 16 sets of ImageNet classes corresponding to 16 basic animal superclasses for our basic animal classification experiments (e.g., see Figs. 1 and S3), adapting and expanding the Restricted ImageNet dataset defined in the Robustness library (Engstrom et al., 2019). The assignment of specific animal classes to each superclass is listed below:

- Dog: classes 151–268
- House Cat: classes 281–285
- Frog: classes 30–32
- Turtle: classes 33–37
- Bird: classes 80–100 and 127–146
- Monkey: classes 369–382
- Fish: classes 0, 1, 389, 391, 392, 393, 394, 395, 396, 397
- Crab: classes 118–121
- Insect: classes 300–320
- Lizard: classes 38–48
- Snake: classes 52–68
- Spider: classes 72–77
- Big Cat: classes 286–293
- Bear: classes 294–297
- Rodent: classes 330, 331, 332, 333, 335, 336, 338
- Antelope: classes 351–353

We excluded certain classes on a case-by-case basis in an attempt to minimize errors due to misunderstanding the animal categories, as opposed to errors of visual perception. For example, we did not include porcupines or beavers in the "rodent" class (as many people may not recognize these as rodents), and we did not include eels in the "fish" class due to the possibility of confusion with

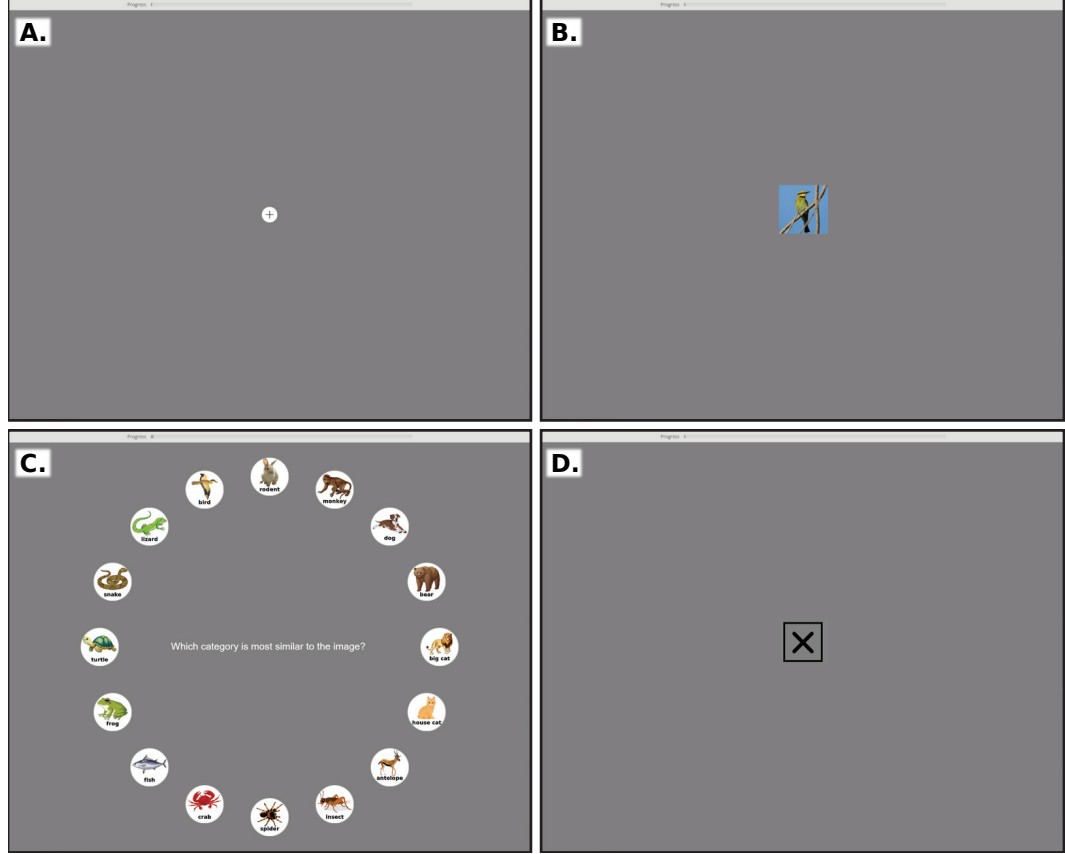

Figure S3: **Task interface for ImageNet animal classification with human participants.** *Subjects classified images among 16 categories. During each trial, the subject clicks the fixation cross (panel **A**) and the image is displayed for 17 milliseconds (panel **B**) with a 200ms presentation of a blank screen immediately before and after. The mouse cursor is hidden during the image presentation. Images are presented such that they subtend approximately 6 degrees of visual angle, with calibration for each participant using a blind-spot calibration procedure. After viewing the image, the participant clicks one of the 16 buttons shown in panel **C**, which are randomly rotated in position every trial, within a 15-second time limit. For incorrect responses, or if 15 seconds elapses without a response, the participant is shown the black X for one second (panel **D**). Otherwise, no explicit feedback is given and the next trial begins immediately. Attention check trials featuring an image of a circle or triangle (see Fig. S4D for an example image) were interspersed with the main trials. For the attention check trials, two of the animal icons in panel **C** were randomly selected to be replaced with circle and triangle icons.*

snakes. We mistakenly classified rabbits and hares as "rodents" given that they were reclassified to the order Lagomorpha in 1912 Chapman & Flux (2008) (we thank the participant who notified us of this).

## S4 DETAILS ON IMAGE CATEGORY LEARNING EXPERIMENTS

Figs. S3 and S4 show the task interfaces for the 16-way animal classification and 4-way image category learning experiments with human participants, respectively. For the learning experiments, the positions of the four buttons used to indicate responses are randomly permuted for each participant.

To minimize biases stemming from any priors induced by the class names, for each participant in the 4-category learning tasks we randomly assign a four-letter, two-syllable alias from the set "Ajax," "Eris," "Leda," and "Tyro." These names are drawn from Greek mythology, and each has four letters, two syllables, two consonants, and two phonetic vowels. We found no evidence that associating

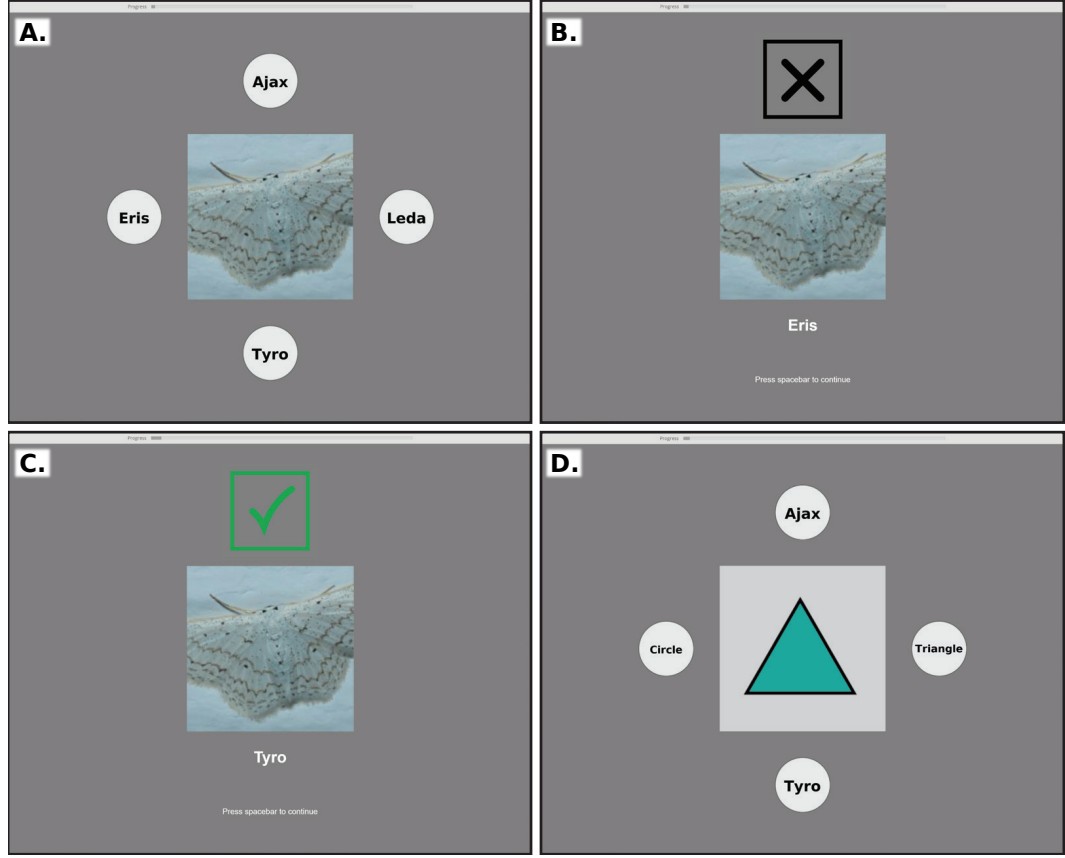

Figure S4: **Task interface for image category learning experiments.** *In the 4-way image category learning experiments (moths, dermoscopy), human subjects learned to classify four types of images that were represented by randomly assigned aliases "Ajax," "Eris," "Leda," and "Tyro." The image was shown for up to 10 seconds, during which the participant could click on one of the four buttons (panel **A**). Participants are shown a black X (1.5 seconds) immediately following an incorrect response or a >10s timeout (panel **B**), or a green check after a correct response (panel **C**). The alias corresponding to the correct class is also displayed on the feedback screen. Panel **D** shows an example of an attention check trial.*

certain categories with certain aliases consistently affected test-phase accuracy (see Figs. S15 and S16).

We used a different approach for the binary histology task that employed the MHIST dataset (Wei et al., 2021), giving "benign hyperplastic polyp" the alias "benign" and sessile serrated adenoma the alias "malignant" (although sessile serrated adenoma is actually a pre-cancerous lesion). The histology task has an interface very similar in appearance to that of the 4-way tasks (as shown in Fig. S4), except that the "benign" and "malignant" buttons always appear on either side of the presented image (in a random order for each participant), and the participant responds by pressing the F key for the left-hand category or J for the right instead of clicking one of the buttons.

## S5 PREDICTING IMAGE DIFFICULTY USING GROUND TRUTH LOGIT OF A ROBUST MODEL, COMPARED WITH PRIOR STATE-OF-THE-ART APPROACHES

In the L-WISE algorithm, we predict the difficulty of each image using its ground truth logit representation ($L_{gt}$) from a robustified ANN such as ResNet-50 (see Figs. 1A, S5, S7, and S14A1,B1,C). We conducted an experiment to compare this ground truth logit score with prior state-of-the-art predictors of image difficulty for humans established by Mayo et al. (2023). We apply logistic re-

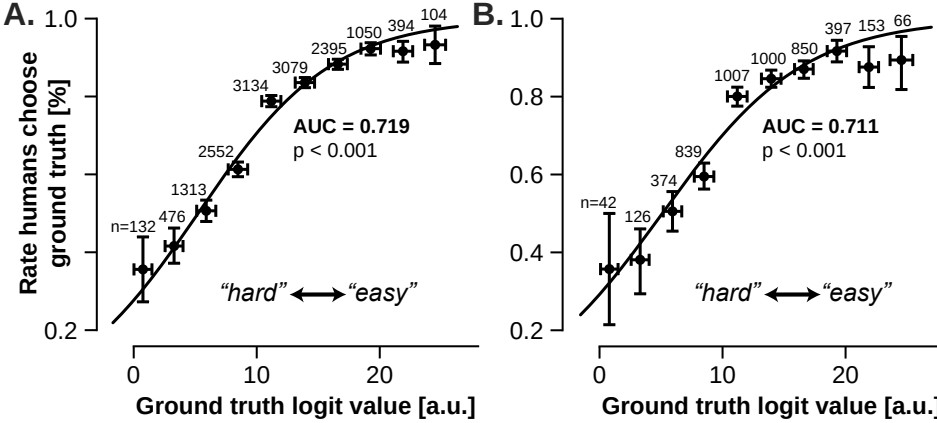

Figure S5: **The observed relationship between robust model ground truth logits and human error rates is not sensitive to trial inclusion criteria.** *For our main analysis regarding predicting ImageNet animal image recognition difficulty illustrated in Fig. 1A, we included all trials with original images but also trials featuring images modified by off-the-shelf control enhancement methods (Adobe Lightroom, CLAHE, and MSRCR), and trials with image perturbations from non-adversarially-trained ANNs (i.e., Vanilla ResNet-50 and CutMix ResNet-50 from Fig. S7). Panel **A** above reproduces Fig. 1A1, while panel **B** replicates the same analysis but strictly including only trials with natural, unmodified images. The numbers above each point indicate how many image trial observations were included in the corresponding bin (the size of the vertical 95% confidence intervals is sensitive to this). AUC=Area Under the Receiver Operating Characteristic Curve, p-values derived from Wald statistic on the logistic regression coefficient for the ground truth logit predictor.*

gression to predict correct v.s. incorrect responses to each (original, unmodified) image across all participants in our 16-way ImageNet animal classification experiment, using (1) c-score (approximated by the epoch during training at which an image is first correctly predicted (Jiang et al., 2021)), (2) prediction depth (earliest layer upon which a linear probe makes the same prediction as the final output (Baldock et al., 2021)), (3) image-level adversarial robustness (minimum magnitude of image perturbation required to change the network's prediction), and (4) ground truth logit from both (A) vanilla and (B) robustified ResNet-50 models (see Fig. S6). C-score, prediction depth, and adversarial robustness are implemented following Mayo et al. (2023). The results show that the ground truth logit from robust ResNet-50 ($L_{gt}$), the metric we use in L-WISE, significantly outperforms all other predictors of image difficulty, including all other metrics combined into one model ("Combined w/o $L_{gt}$" in Fig. S6). Furthermore, combining all other metrics with $L_{gt}$ does not improve performance beyond $L_{gt}$ alone. We also find that using a robustified model rather than a "vanilla" model to generate each metric greatly improves the predictivity of $L_{gt}$ and (marginally) adversarial robustness, but not of the c-score or prediction depth.

## S6 COMPARISON OF DIFFERENT ANN GUIDE MODELS FOR DIFFICULTY PREDICTION AND IMAGE ENHANCEMENT

Our main results use robustified ResNet-50 as a guide model for generating image perturbations. To evaluate the importance of the choice of guide model, we compared the accuracy of difficulty prediction (Fig. S7) and the effects of enhancement with $\epsilon = 20$ (Fig. S9) using 6 different guide models in the 16-way animal classification task. The results show that setting $\epsilon$ to a value of 3 during adversarial training of ResNet-50 yields more accurate difficulty predictions and more effective perturbations than $\epsilon = 1$ or $\epsilon = 10$ training (consistent with disruption modulation results in Gaziv et al. (2023)), while perturbations guided by a non-adversarially-trained "vanilla" model have negligible effects. Model accuracy seems to be less important than robustness, at least for difficulty prediction: ground truth logits from ResNet-50 models at early epochs of adversarial training on ImageNet, which have lower image classification accuracy than models at later epochs, can still yield accurate image difficulty predictions (Fig. S8A). Models at early training epochs may also be able to generate high-quality image enhancement perturbations (Fig. S8B), although we did not study this in our experiments with human participants. Although training ResNet-50 with CutMix improves its

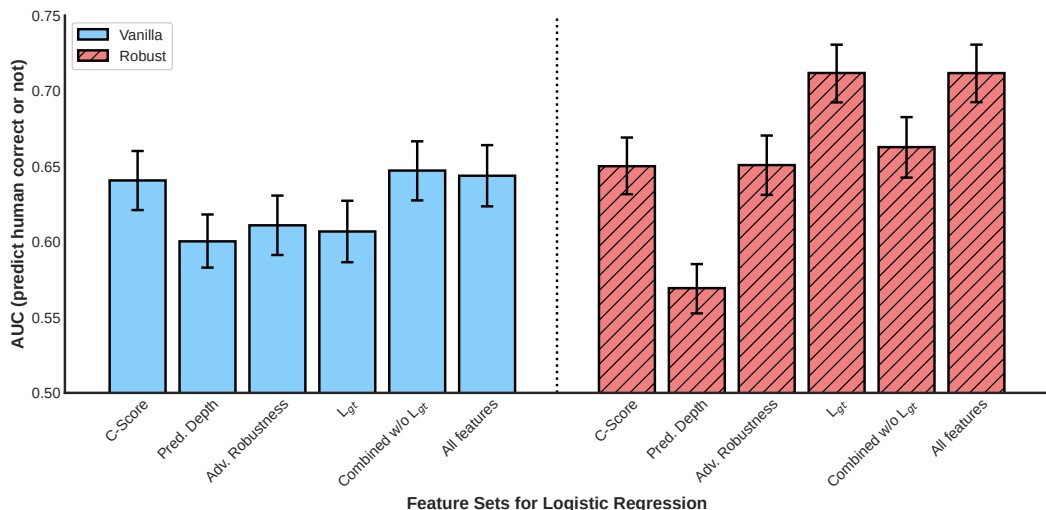

Figure S6: **Robustified ground truth logit is a state-of-the-art predictor of image difficulty for humans, outperforming the c-score, prediction depth, and adversarial epsilon of both vanilla and robustified models.** *AUC estimates are based on fitted logistic regression models using one or more features listed under each bar, with stratified 500-fold cross-validation. Error bars are 95% confidence intervals for the mean from 10,000 bootstrap replicates. The chance level is AUC=0.5.*

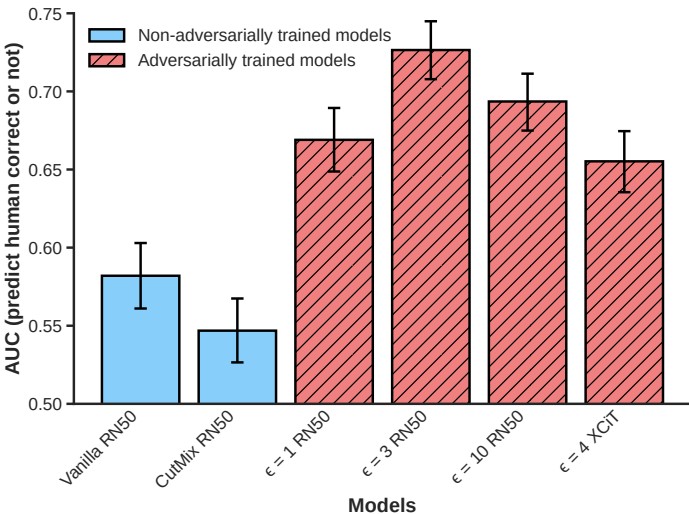

Figure S7: **Accuracy of image difficulty prediction using ground truth logits from different model types.** *AUC estimates and 95% confidence interval error bars are generated by the same procedure as in Fig. S6 - however, results are not directly comparable between the two figures as different ANN training runs were used for consistency within each experiment. RN50 = ResNet-50, and ε values in the labels for each bar show the magnitude of the adversarial perturbations during adversarial training.*

robustness to adversarial perturbations (Yun et al., 2019), CutMix-ResNet-50 does not outperform vanilla ResNet-50 in image difficulty prediction and perturbations using it as a guide model do not significantly increase accuracy beyond that on original images. In addition to ResNet-50, we tested the difficulty prediction and image enhancement capabilities of an adversarially trained vision transformer model, the Cross-Covariance Image Transformer (XCiT) (Ali et al., 2021). Debenedetti et al. (2023) showed that the XCiT architecture is more suitable for adversarial training than the original vision transformer. XCiT generates reasonably accurate image difficulty predictions (on par with the previous state-of-the-art) and generates image perturbations that increase human categorization

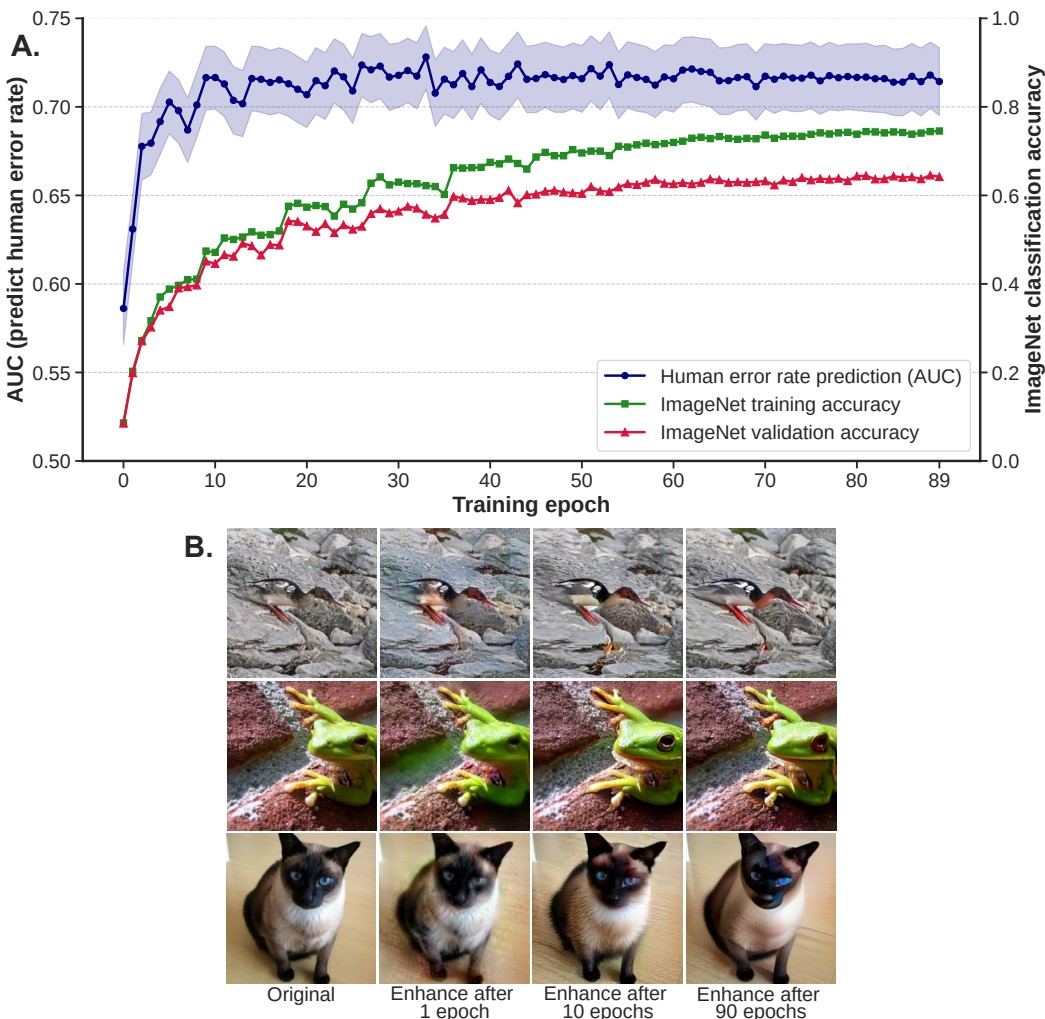

Figure S8: **Robustified model accuracy weakly affects the relationship between ground truth logit and human image difficulty.** *To generate this figure, we retained ResNet-50 model checkpoints from immediately after each of 90 epochs of adversarial training on ImageNet-1K from scratch ($\epsilon = 3$, batch size = 256, initial learning rate of 0.1 decreases by half every 9 epochs). In panel **A**, AUC estimates of logistic regression models predicting human trial response correctness were generated by the same procedure as in Appendix Figs. S6-S7, using ground truth logits generated by each of the epoch checkpoints. The shaded area denotes 95% confidence intervals around each mean AUC from 10,000 bootstrap replicates. The training set and validation set accuracy at 1000-way ImageNet-1K classification, on non-perturbed images, is also plotted by training epoch. We observe that the AUC of human error rate prediction stops increasing relatively early during training, well before the training/validation accuracy on the ImageNet classification task is saturated. Panel **B** shows example images with $\epsilon = 20$ enhancements generated by checkpoints at various stages of the training process.*

accuracy by a comparable degree to robustified ResNet-50. For the experiments in Figs. S7 and S9, we used pretrained guide models provided by Gaziv et al. (2023) (Vanilla, $\epsilon = 1$, $\epsilon = 3$, and $\epsilon = 10$ ResNet-50 models), Yun et al. (2019) (CutMix ResNet-50), and Debenedetti et al. (2023) ($\epsilon = 4$ XCiT). Examples of images enhanced by each of these guide models with $\epsilon = 20$ are displayed in Fig. S10.

Image perturbations generated with vision transformer models (such as XCiT) typically include grid-like artifacts related to the image patch/grid structure of these models (Dosovitskiy et al., 2020). To mitigate grid artifacts during each step of generating image perturbations with XCiT, we calculated gradients with respect to each pixel value by averaging across ten randomly translated, resized (fol-

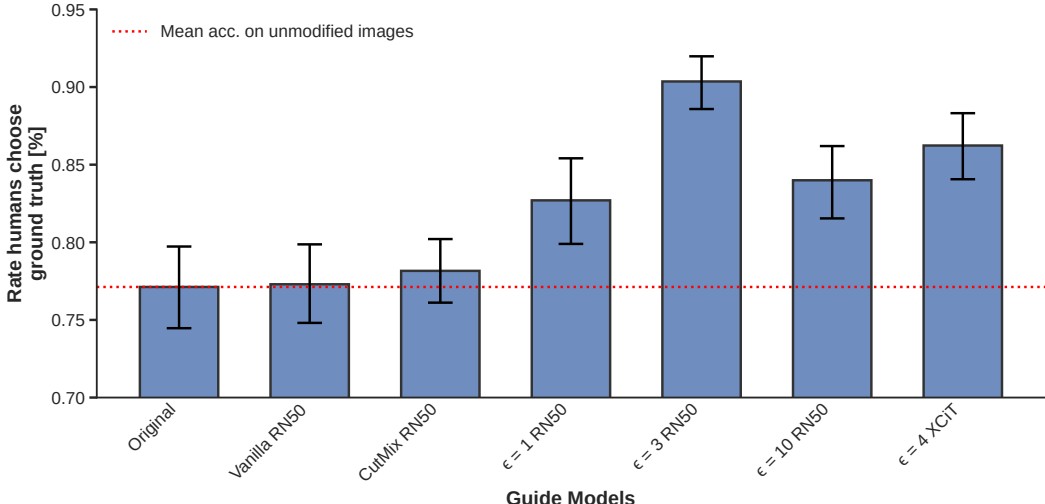

Figure S9: **Effectiveness of image category enhancement across different guide model types.** *Each bar shows the mean and 95% confidence interval (by bootstrap) of the rate at which humans choose the original ground truth label, in a 16-way basic animal classification task using ImageNet images. The "Original" bar shows accuracy for unmodified images, and other bars show accuracy of the same participants on images enhanced ($\epsilon = 20$) using gradients from the corresponding guide model. RN50 = ResNet-50, and $\epsilon$ values in the labels for each bar show the magnitude of the adversarial perturbations during adversarial training.*

lowed by cropping/padding), color-jittered, and randomly cut-out "views" of each image, a strategy inspired by Ganz & Elad (2024) that extends DiffAugment (Zhao et al., 2020).

## S7 ABLATION STUDY ON LOGIT MAXIMIZATION APPROACH TO ENHANCEMENT

As a limited ablation study on our approach to image enhancement, we conducted an additional 16-way ImageNet animal classification experiment with 20 human participants. This experiment was mostly identical to the 16-way animal classification experiment described in the main text, except there were 6 image conditions instead of 9. Half of the trials used images from the ImageNet validation set (as in the main experiment), and the other half from the training set. Within each training/validation split, one third of the trials were original, unmodified images, one-third were enhanced by maximizing the ground truth logit with $\ell_2$ pixel budget $\epsilon = 10$, and one-third were enhanced by minimizing the cross-entropy loss with $\epsilon = 10$. The results of this experiment are summarized in Fig. S11A. We hypothesized that logit-based enhancement would provide superior results, particularly for images that started off with low cross-entropy loss. We further hypothesized that enhancements would be less effective for training images due to overfitting of the guide model on them. The results show that logit maximization is effective on both training and validation images, and induces a larger increase in accuracy for a given pixel budget $\epsilon$ than cross-entropy minimization. Indeed, cross-entropy minimization significantly increased accuracy only for validation images and not for training images. Unexpectedly, participants were more accurate on original, unmodified training set images than on original, unmodified validation set images. According to Russakovsky et al. (2015), the ImageNet ILSVRC 2012 validation set was collected using the same methodology as the training set, but at a later time. It is therefore plausible that the images and labels in the validation set are drawn from a slightly different distribution than those in the training set, resulting in this accuracy discrepancy.

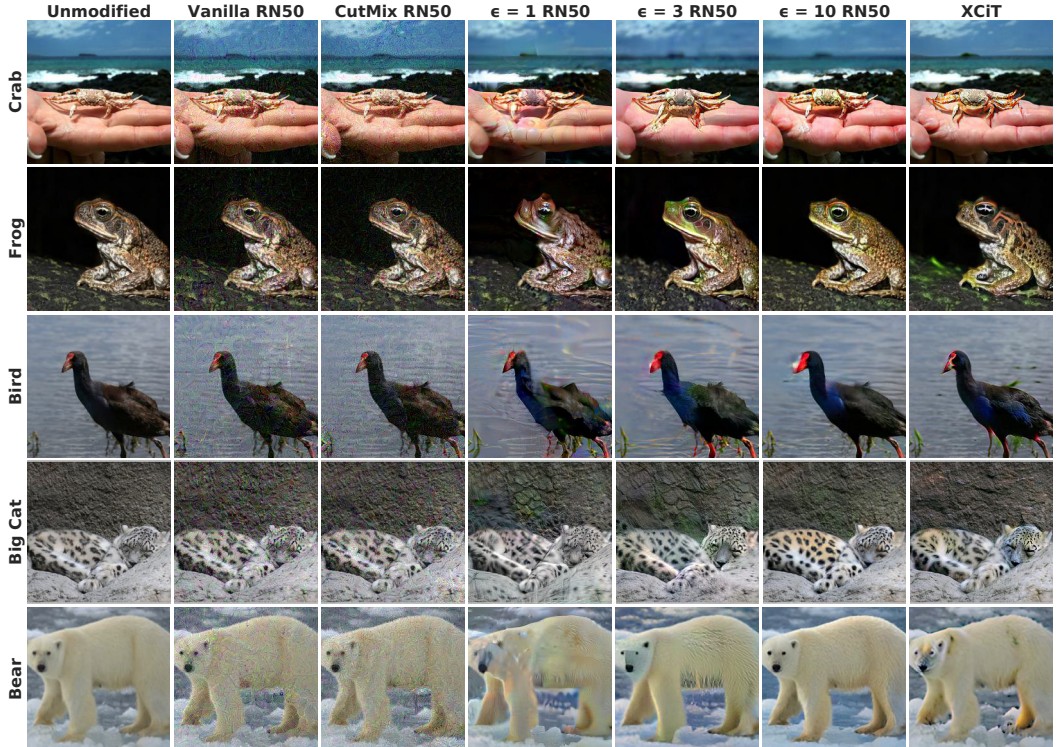

Figure S10: **Meaningful perturbations require robust models, and are possible with CNN and vision transformer architectures.** *Each row shows an image from ImageNet (original on the far left) enhanced with $\epsilon = 20$ by different guide models. A quantitative comparison of different models' perturbation efficacies with regards to improving human classification accuracy can be found in Appendix Fig. S9.*

## S8 ADDITIONAL RESULTS FROM IMAGE CATEGORY LEARNING EXPERIMENTS

Fig. 3 in the main text shows learning curves (mean smoothed accuracy by condition as a function of trial number), and schedules for image difficulty selection and enhancement $\epsilon$, for the moth photograph task: similar plots are shown for the dermoscopy task in Fig. S12 and the histology task in Fig. S13. Panels H-M of Fig. S12 show the results of an early pilot experiment that used image enhancement in isolation (no difficulty selection), in which we suspect the perturbation magnitude $\epsilon$ was set too high causing participants to learn exaggerated features and fail to generalize to natural images with subtler features. This prompted us to switch to the $\epsilon$ schedule we used for our main learning experiments, which starts at $\epsilon = 8$ instead of $\epsilon = 20$. Panel C of Fig. S13 shows the relationship between the ground truth logit from robust ResNet-50 model and how many of the 7 expert annotators of the MHIST histology dataset Wei et al. (2021) agreed on the same category label. On average, the model is more "confident" in its predictions (higher ground truth logit) on images where experts are more in agreement with each other.

In addition to the agreement of expert MHIST annotators, the ground truth logit successfully predicts the proportion of human participants who select the correct ground truth label across all tasks we tested. Difficulty prediction results from the 16-way ImageNet task are shown in Fig. 1 in the main text, and from the moth photograph, dermoscopy, and histology tasks in Fig. S14 (Panels A1, B1, and C). For this analysis in the non-ImageNet tasks, we rely on test-phase data from control group participants who had just learned the tasks in question.

We can also attempt to measure the extent to which images with higher levels of enhancement are easier for novice participants to recognize during the learning tasks (Fig. S14A2,B2). This analysis is limited to the first training trial blocks in the "ET (shuffled)" participant group in the ablation

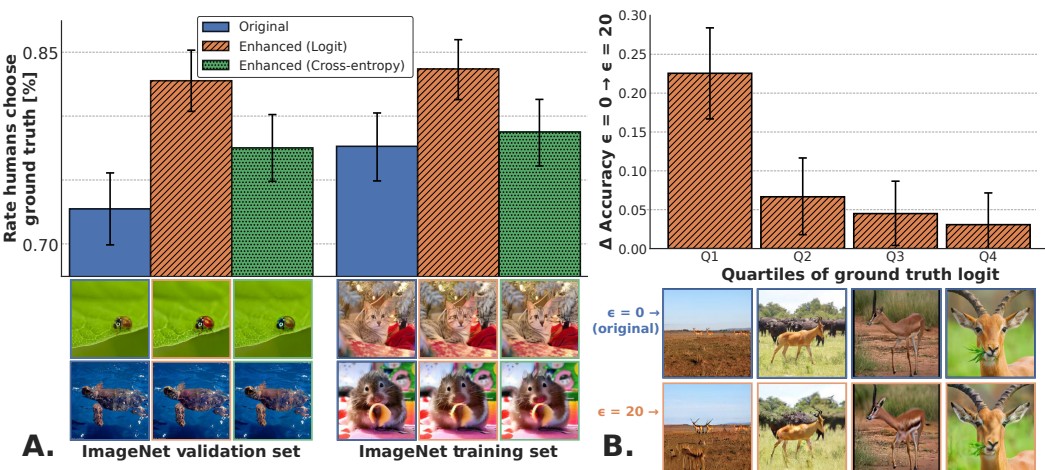

Figure S11: **Ablation results for image enhancement with ImageNet images.** *Logit maximization enhancement is effective for images used to train the robustified CNN used as a guide model, and also for held-out validation images (panel **A**). Logit-max enhancement is more efficient at increasing human accuracy within a given pixel budget ($\epsilon = 10$) than enhancement by cross-entropy minimization (panel **A**). The efficacy of logit-max enhancement depends on the difficulty of the original image as estimated by the starting ground truth logit (panel **B**). In the bar plot of panel **B**, images were assigned to 4 quadrants based on their ground truth logit values, and for each quadrant the mean difference in accuracy was calculated between original, unmodified images and images enhanced with $\epsilon = 10$ (using data from the main 16-way animal classification experiment). The images below each bar illustrate an example image from the category "antelope" drawn from the corresponding difficulty quadrant. All error bars are 95% confidence intervals for the mean from 10,000 bootstrap replicates.*

study (main-text Table 1), the only group that viewed enhanced images without $\epsilon$ monotonically decreasing over time. Note that there were 6 discrete $\epsilon$ values (1 per block in the non-shuffled ET condition), and the analysis is complicated by the fact that participants were still learning the task when they made the responses underlying these plots. We are also unable to compare with $\epsilon = 0$ unmodified images because participants did not view new unmodified images in the corresponding training blocks. There are no results here for the histology task because the ablation study was conducted only for moth photos and dermoscopy images. In both the moth task and the dermoscopy task, participants respond with the original, correct category label statistically significantly more often when viewing images enhanced with greater $\epsilon$ (Fig. S14B1,B2).

To evaluate whether L-WISE has differential effects on human image category learning depending on the image class, we record test-phase precision and recall for each class among L-WISE and control groups in Table S1. The same data are visualized in Fig. 4B. Our experiments are statistically underpowered to detect class-specific differences in performance (as opposed to aggregated performance) - however, we can observe in a coarse sense that the sample means of precision and recall are numerically higher in the L-WISE group across all classes in all tasks. This suggests that overall accuracy improvements attributed to L-WISE are distributed among the various image classes, rather than being the result of isolated improvements in the detection of a subset of classes.

## S9 PARTICIPANT DROPOUT RATES ARE LOWER WHEN L-WISE ASSISTANCE IS PROVIDED

On the Prolific platform where we ran our experiments, participants can choose to withdraw from studies partway through if they no longer wish to participate (this is called "returning" a study in the Prolific interface). For the moth photograph and dermoscopy image category learning tasks, participants who received L-WISE assistance in full or partially ablated form (see Table 1) were less likely to withdraw.

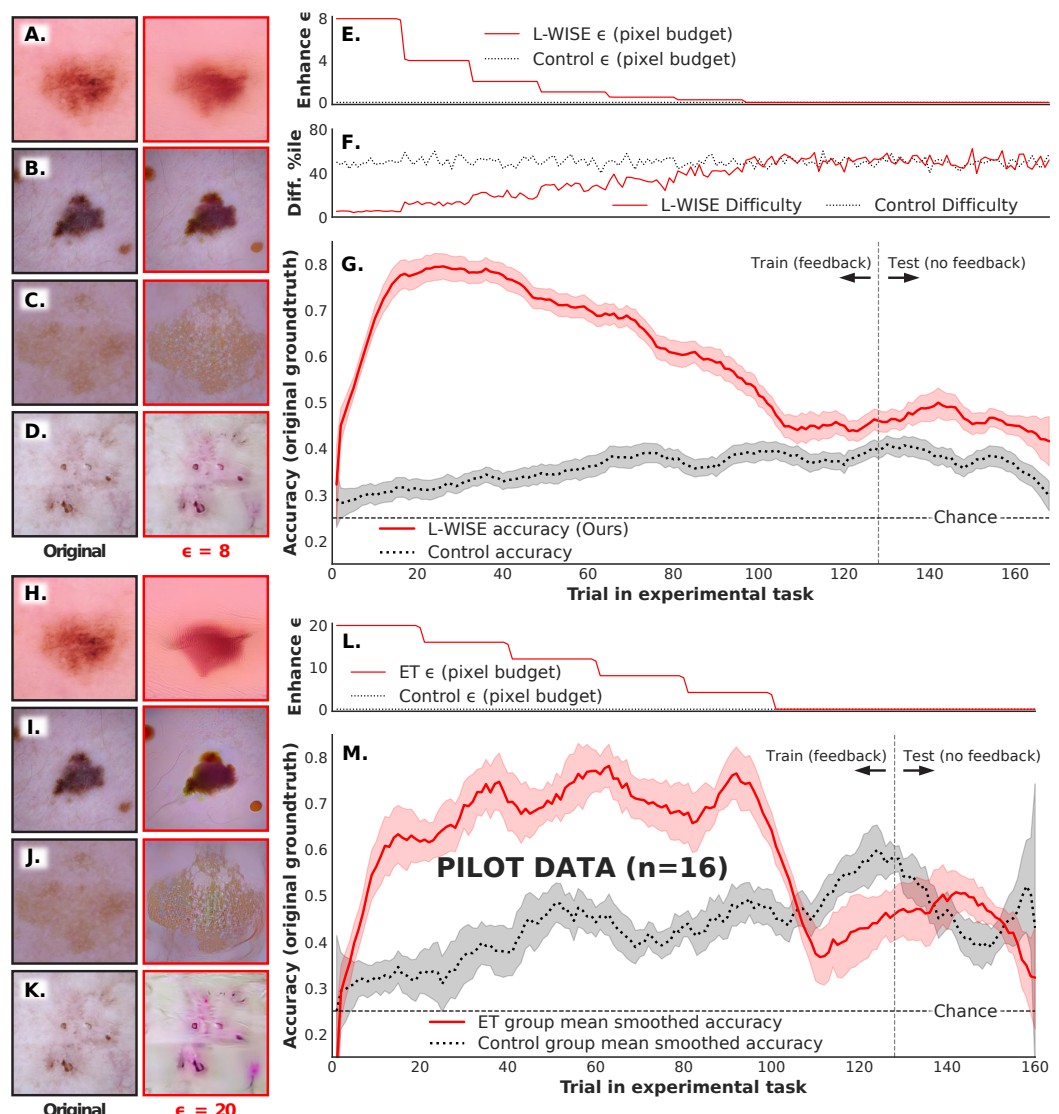

Figure S12: **Plots showing the accuracy trajectory of human participants throughout training/testing in the main dermoscopy learning experiment (panels A-G) and a preceding pilot experiment after which the epsilon tapering schedule was adjusted (panels H-M).** *All conventions are identical to main-text Fig. 3. There is a statistically significant difference between the test-phase performance of the L-WISE participants and that of the control participants ($\chi^2(1)$ test, $p < 0.001$) in panel G but not for the pilot experiment in panel M. Notably, the last portion of the training phase does not feature any image enhancements (see Fig. 2F): we suspect that this is the reason for the sudden decline in accuracy in the enhancement group of the pilot experiment (M).*

Nine participants withdrew from the moth photograph category learning experiment. Among them, six had been assigned to the control group, one to the "enhancement taper" group, one to the "difficulty selection" group, and one to the full L-WISE group. We can calculate the probability of $d = 6$ or more participants among the $n = 9$ who withdrew being from the control group, under the null hypothesis that the probability of withdrawal is independent of group assignment, using the binomial distribution via Equation 3 below (where $p$ is the probability of being assigned to the control group). Equation 3 evaluates here to a probability of $p = 0.02$, indicating that participants who withdrew were significantly more likely to have been assigned to the control group than would be expected if L-WISE assistance had no impact on the probability of withdrawal.

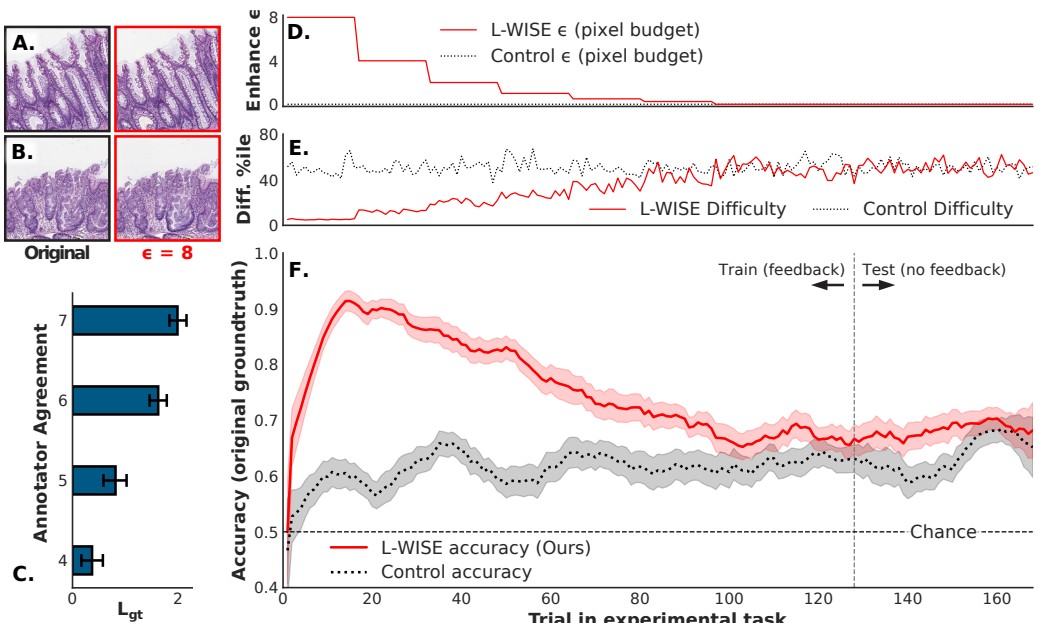

Figure S13: **Plot showing the accuracy trajectory throughout training/testing of human participants in the histology learning experiment.** *All conventions follow Fig. 3. Panel C shows the association between agreement among the 7 expert pathologist annotators of the MHIST dataset (Wei et al., 2021) and the ground truth logit score of each image from a robustified ResNet-50. Possible values of annotator agreement are 4, 5, 6, or 7 of the annotators agreeing with each other (3 and below switches the "ground truth" category). Error bars are 95% confidence intervals for the mean from 10,000 bootstrap samples.*

$$P(X \geq d) = 1 - P(X \leq d - 1) = 1 - \sum_{k=0}^{d-1} \binom{n}{k} p^k (1-p)^{n-k} \tag{3}$$

Similarly, in the dermoscopy category learning experiment, 13 participants withdrew, of whom 6 were from the control group. In this case, Equation 3 evaluates to a probability of $p = 0.041$, again indicating that participants in the control group withdrew at a significantly higher-than-expected rate. Furthermore, 4 more of the 13 withdrawals were from the "Enhancement Taper (shuffled)" group, which had test-phase accuracy indistinguishable from the control group (see Table 1). None of the withdrawals from the dermoscopy experiment were from the full L-WISE group.

Overall, these results show that participants were more likely to withdraw from the study when they did not receive assistance from L-WISE, perhaps reflecting the difficult nature of the moth photograph and dermoscopy image tasks at baseline. None of the participants withdrew from the histology image experiment, precluding a similar analysis.

## S10 Notes on "hallucinations" in enhanced images

To support our approach to assisting human learners, we demonstrate the ability to enhance category percepts in images using low-norm perturbations. Previous work by Gaziv et al. (2023) showed that an image from one category can be perturbed in a targeted way such that a human perceives it to belong to a different category. Features introduced by these disruptive perturbations could be described as "hallucinations:" perceptions (by the model and the human viewer) of objects that are not present in the camera's view. Our image enhancement approach is a special case in the wider realm of categorically targeted image modulation, in which maximization of the ground truth logit perturbs the image such that it becomes a stronger and/or less ambiguous example of its class according to the model's judgement. Do these perturbations accentuate features that are already present such that they are easier for humans to perceive under challenging conditions, or do they improve hu-

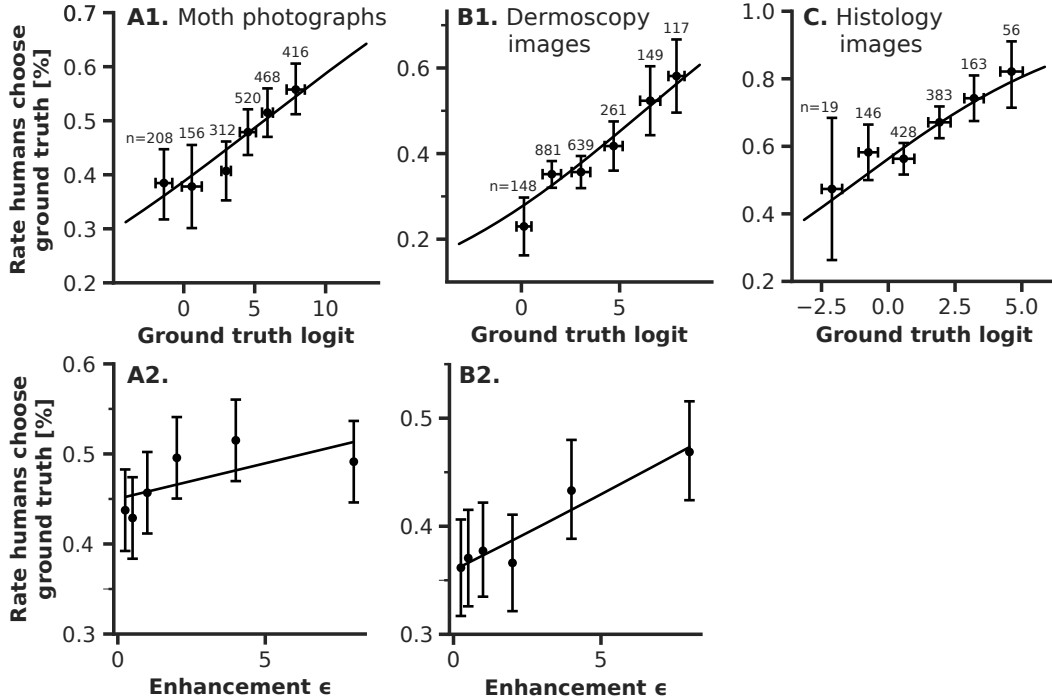

Figure S14: **Difficulty prediction and image enhancemement are effective across image domains.** *Panels A1, B1, and C show the relationship between the ground truth logit from a fine-tuned robustified ResNet-50 model and the rate at which human participants (from the control groups) choose the ground truth label during the test phase following a training phase in which they had just attempted to learn the task. Images are binned by ground truth logit to produce the scatter plots, with the number of total trials listed for each bin. Vertical error bars are 95% confidence intervals by bootstrap, and horizontal error bars show the standard deviation within each bin. The curved lines illustrate fitted logistic regression models. All logistic regression models had statistically significant coefficients for ground truth logit ($p < 0.001$ from Wald statistic). Panels A2 and B2 show the relationship between enhancement $\epsilon$ and the rate at which humans choose the ground truth category. This analysis is limited to the first training trial blocks in the "ET (shuffled)" participant group in the ablation study (main-text Table 1), the only group that viewed enhanced images without $\epsilon$ monotonically decreasing over time. The logistic regression coefficient for $\epsilon$ was statistically significant ($p < 0.05$ from Wald statistic) for both moth photographs (A2) and dermoscopy images (B2).*

man accuracy by hallucinating new features associated with the target category? Subjectively, both phenomena seem to occur: panel **B2** in Fig. 1 appears to show bolder contrasts and exaggerated features in class-relevant regions of the perturbed images. Panel **B** (image farthest to the left) in Appendix Fig. S11, however, shows a clear example of hallucination, where a semblance of an entire additional "antelope" appears in the foreground of the image. This distinction may be important for education-oriented applications of our enhancement approach, as hallucinations could plausibly impart potentially misleading information to the learner. On the other hand, it is possible that hallucinated features can impart useful and therefore desirable representations of the ground truth class despite departures from a natural image distribution.

## S11 PARTICIPANT RECRUITMENT AND DEMOGRAPHICS

We recruited a grand total of 521 participants via the online platform Prolific. All participants lived in the United States and were fluent in English (as determined by Prolific). Each participant was eligible to complete each learning experiment only once, to avoid collecting data from participants already familiar with a given task.

Our decision regarding the number of participants to recruit for each learning task experimental group (targeting 30 on average) was intended to exceed the requirements of a simple power anal-

| | Precision | | Recall | |
|---|---|---|---|---|
| | Control | L-WISE | Control | L-WISE |
| **Moth photos** | | | | |
| *seriata* | 0.46 (0.39–0.52) | **0.52** (0.45–0.59) | 0.43 (0.37–0.50) | **0.63** (0.55–0.71) |
| *tacturata* | 0.45 (0.40–0.50) | **0.63** (0.55–0.71) | 0.54 (0.47–0.61) | **0.68** (0.60–0.76) |
| *biselata* | 0.35 (0.30–0.41) | **0.41** (0.32–0.50) | 0.36 (0.31–0.42) | **0.42** (0.35–0.50) |
| *aversata* | 0.50 (0.44–0.56) | **0.58** (0.50–0.66) | 0.57 (0.50–0.63) | **0.68** (0.59–0.77) |
| **Dermoscopy** | | | | |
| Benign mole | 0.38 (0.33–0.43) | **0.42** (0.36–0.48) | 0.43 (0.38–0.48) | **0.47** (0.40–0.54) |
| Melanoma | 0.33 (0.29–0.38) | **0.39** (0.35–0.44) | 0.37 (0.31–0.42) | **0.42** (0.37–0.47) |
| BCC | 0.41 (0.36–0.46) | **0.54** (0.46–0.61) | 0.41 (0.36–0.47) | **0.56** (0.48–0.63) |
| Benign keratosis | 0.26 (0.22–0.30) | **0.39** (0.34–0.43) | 0.30 (0.25–0.35) | **0.43** (0.36–0.49) |
| **Histology** | | | | |
| SSL (malignant) | 0.58 (0.54–0.62) | **0.60** (0.56–0.64) | 0.66 (0.61–0.71) | **0.70** (0.66–0.75) |
| HP (benign) | 0.59 (0.55–0.64) | **0.61** (0.56–0.67) | 0.61 (0.56–0.66) | **0.66** (0.62–0.70) |

Table S1: **L-WISE improves test-phase precision and recall across all image classes in three image category learning tasks.** *BCC=basal cell carcinoma, SSL=sessile serrated adenoma, and HP=hyperplastic polyp. In parentheses are 95% confidence intervals for the mean from 10,000 bootstrap replicates, resampling from participant-wise precision and recall values.*

ysis we conducted following pilot experiments. Pilot experiments showed differences in test-time accuracy between control and either enhancement taper (equivalent to ET in main-text Table 1) or difficulty selection (equivalent to DS in Table 1) participants to be roughly 10%, with a standard deviation in accuracy of roughly 10% in each group.

$$H_0 : \mu_1 - \mu_2 = 0$$
$$H_1 : \mu_1 - \mu_2 \neq 0$$

Given:

$$\delta = 0.1 \text{ (estimated mean difference)}$$
$$\sigma = 0.1 \text{ (estimated standard deviation)}$$
$$\alpha = 0.05 \text{ (significance level)}$$
$$1 - \beta = 0.8 \text{ (power)}$$

$$\text{Estimated effect size } d = \frac{\delta}{\sigma} = 1.0$$

$$\text{Required sample size per group: } n = 2(z_{1-\alpha/2} + z_{1-\beta})^2/d^2$$
$$= 2(1.96 + 0.84)^2/1^2$$
$$\approx 16 \text{ subjects per group at minimum}$$

We provide a demographic breakdown of the participants in our study, aggregated across experiments, in Table S2. Some participants took part in more than one of the experiments, but are only counted once in the table.

| | |
|---|---|
| Total participants | 521 |
| Pts. w/ demographic data | 519 (99.6%) |
| Age | |
|   Mean (SD) | 36.6 (11.9) years |
|   Range | 18-83 years |
| Sex | |
|   Female | 289 (55.7%) |
|   Male | 227 (43.7%) |
|   Not specified | 3 (0.6%) |
| Ethnicity | |
|   White | 338 (65.1%) |
|   Black | 54 (10.4%) |
|   Asian | 50 (9.6%) |
|   Mixed | 44 (8.5%) |
|   Other | 23 (4.4%) |
|   Not specified | 10 (1.9%) |

Table S2: Demographic characteristics of study participants, aggregated across all experiments.

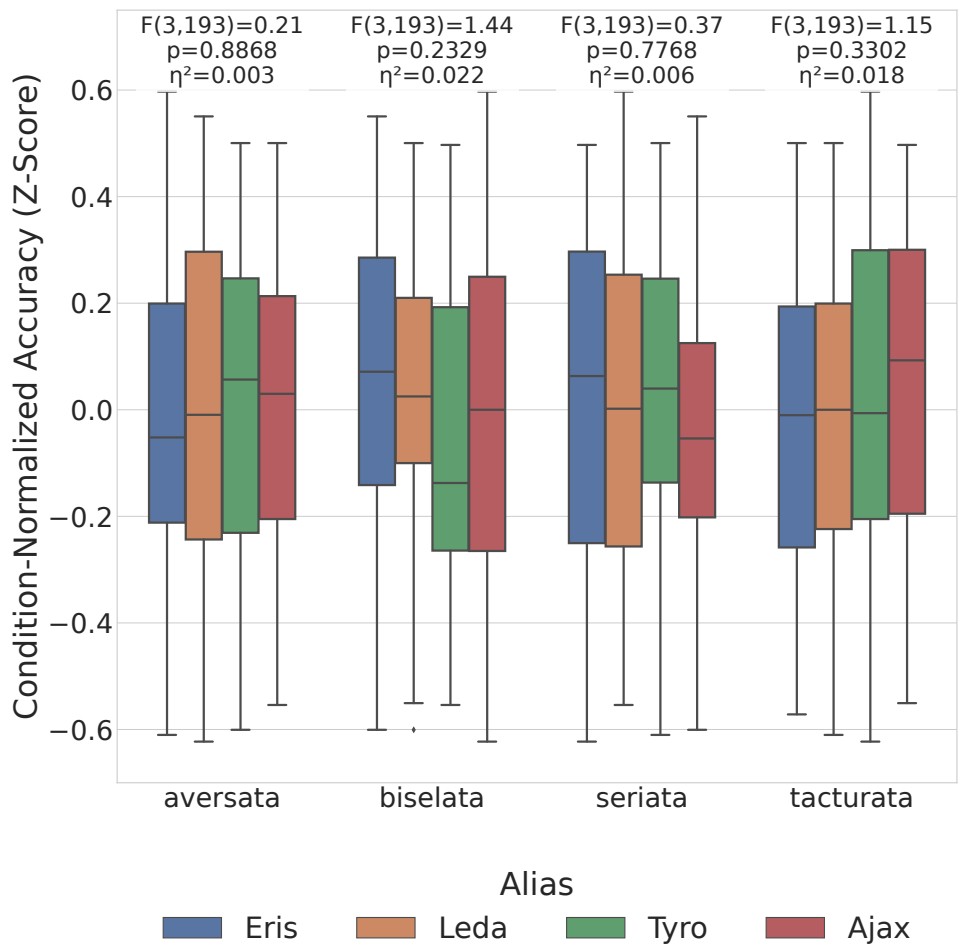

Figure S15: **Randomized assignment of aliases "Leda," "Ajax," "Eris," and "Tyro" had minimal impact on test-phase accuracy in the moth species classification task.** *Each group of four boxplots shows the relative effects of assigning each alias to a specific class from the moth classification experiment. Each individual boxplot indicates the distribution of participant-wise test-phase accuracy z-scores (normalized with mean and standard deviation within each condition separately) among participants with mapping of a specific alias onto a specific class - for example, the left-most boxplot within the right-most group describes the accuracy of participants who saw images of the moth species Idaea tacturata labelled as "Eris." There is no evidence from one-way ANOVA that the random assignment of aliases to classes influences test-phase performance.*

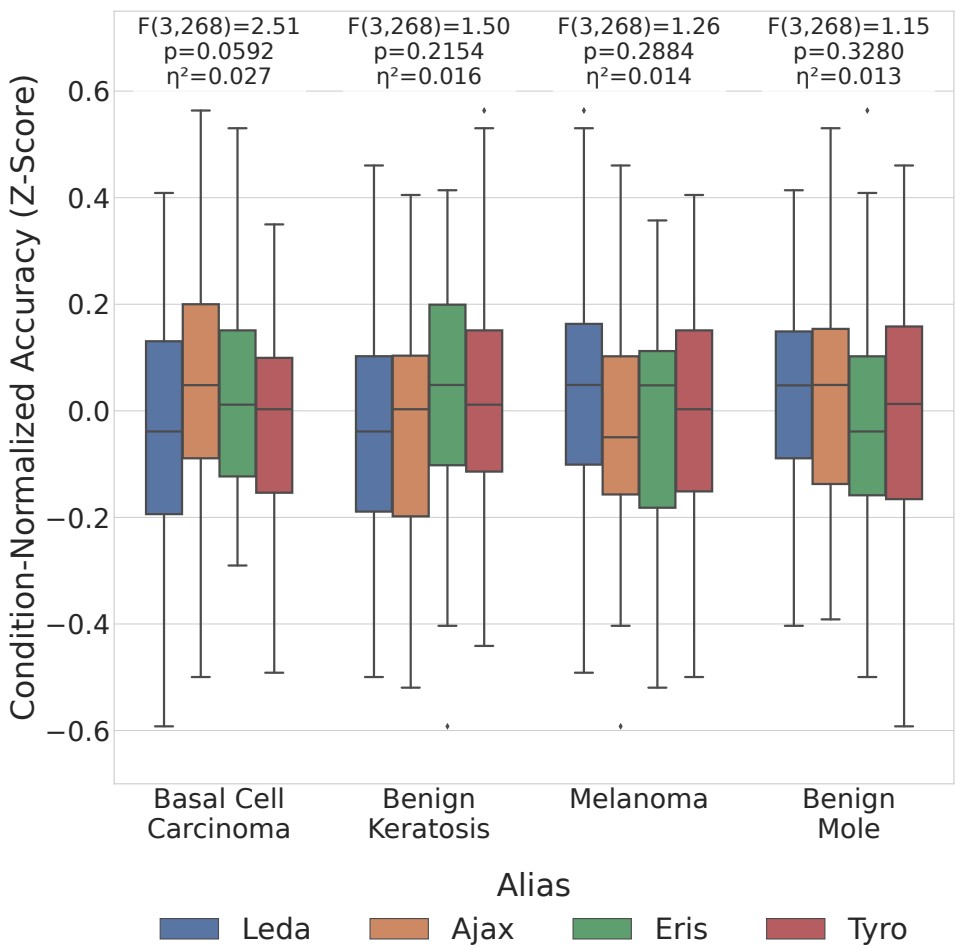

Figure S16: **Randomized assignment of aliases "Leda," "Ajax," "Eris," and "Tyro" had minimal impact on test-phase accuracy in the dermoscopy task.** *After correcting for multiple comparisons, there is no evidence that the random assignment of aliases to classes affects task performance. See also Fig. S15.*

