# OpenReview forum: "L-WISE: Boosting Human Visual Category Learning Through Model-Based Image Selection and Enhancement"
_ICLR.cc/2025/Conference — ICLR 2025 Poster_

### Official Review · Reviewer_nJDg · 2024-11-02

**Soundness:** 2
**Presentation:** 2
**Contribution:** 2
**Rating:** 6
**Confidence:** 3

**Summary:**

This paper proposes a method to enhance human visual learning by designing a model-based selection and enhancement algorithm to improve classification accuracy during testing. First, the authors select images to present to novice learners based on a model’s estimated recognition difficulty for each image. Next, they apply image perturbations intended to aid recognition for novice learners. The authors conduct experiment on the three benchmark datasets, including the natural image and the clinical image to verify the effectiveness of their proposal.

**Strengths:**

1. This paper is well-motivated, and a decent amount of technical details are given.
2. The idea of improving the categorization performance of the novice learner by leveraging the capacity of the robustified artificial neural network is both interesting and practical.
3. The reported improvement in novice learners' performance is notable, with gains in both test accuracy and reduced training time.

**Weaknesses:**

1. The establishment of the empirical observations is somewhat unconvincing. Do these observations hold in more complex classification tasks or when applied to medical imaging?
2. The related work section lacks discussion of both the machine teaching and human-machine vision alignment methods.
3. The size of the particants is somewhat small.
4. The perception of enhanced images may be altered due to perturbations.

**Questions:**

1. The empirical observations are derived from a 16-way animal categorization task on natural images, which seems somewhat simplistic. It would be valuable to examine how these observations hold up in more complex categorization tasks or with different types of images, especially medical images. Given the typically limited availability of medical images, the proposed method could have promising applications in the medical imaging field.

2. Beyond the empirical observations, is there any physiological insight or analysis on why the proposed model-based selection and enhancement method could improve novice learners’ performance in categorization tasks?

3. The authors do not discuss the related machine teaching literature. An in-depth comparison with machine teaching methods, particularly with "Teaching Categories to Human Learners with Visual Explanations" (CVPR 2018), would be valuable. This work similarly considers image difficulty; an introduction and comparison with it are beneficial.

4. The authors should also discuss the connection to human-machine vision alignment methods, such as "Harmonizing the Object Recognition Strategies of Deep Neural Networks with Humans" (NeurIPS 2022).

5. The sample size of participants is relatively small, and expanding the participant pool is recommended to enhance the reliability of the results; also, recruiting participants from diverse backgrounds would improve the generalizability of the findings. If expanding the participant pool is impractical due to time or budget constraints, performing a power analysis or discussing effect sizes could help strengthen the reliability of the analysis.

6. When using perturbations to enhance images, how does the method ensure that essential image details remain unchanged, particularly when using a large ϵ (e.g., 20)? This concern is especially pertinent for medical images, where even slight pixel changes may alter critical information. I recommend involving medical experts to review the enhanced images or using quantitative similarity measures (such as SSIM or FSIM) to verify that essential details are preserved.

---

> ### Author Response · Authors · 2024-11-27
> **Response to feedback from reviewer nJDg (part 1)**
>
> We thank the reviewer for their time and effort in reviewing our submission, and for their insightful feedback, which has led to improvements in our manuscript.
>
> **Weaknesses**
>
> 1. _The establishment of the empirical observations is somewhat unconvincing. Do these observations hold in more complex classification tasks or when applied to medical imaging?_
>
> We assume that the reviewer is referring to panels A1 and B1 of Figure 1, which show that robust ResNet-50 models can predict image difficulty and generate perturbations that increase image categorization accuracy on a 16-way animal classification task derived from ImageNet. To address the reviewer’s concerns, we have added Figure S11 to the Appendix, extending image difficulty prediction and image enhancement observations to the three fine-grained image category learning experiments. Paragraphs 2 and 3 of Appendix Section S8, which are newly added, read as follows:
>
> “
>
> In addition to the agreement of expert MHIST annotators, the ground truth logit successfully predicts the proportion of human participants who select the correct ground truth label across all tasks we tested. Difficulty prediction results from the 16-way ImageNet task are shown in Fig. 1 in the main text, and from the moth photograph, dermoscopy, and histology tasks in Fig. S11 (Panels A1, B1, and C). For the non-ImageNet tasks, we rely on test-phase data from control group participants who had just learned the tasks in question.
>
> We can also attempt to measure the extent to which images with higher levels of enhancement are easier for novice participants to recognize during the learning tasks (Fig. S11A2,B2). This analysis is limited to the first training trial blocks in the “ET (shuffled)” participant group in the ablation study (main-text Table 1), the only group that viewed enhanced images without monotonically decreasing ϵ. Note that there were 6 discrete ϵ values (1 per block in the non-shuffled ET condition), and the analysis is complicated by the fact that participants were still learning the task when they made the responses underlying these plots. We are also unable to compare with ϵ = 0 unmodified images because participants did not view new unmodified images in the corresponding training blocks. There are no results here for the histology task because the ablation study was conducted only for moth photos and dermoscopy images. In the dermoscopy task, participants respond with the original, correct category label statistically significantly more often when viewing images enhanced with greater ϵ (Fig. S11B2). A similar trend was not statistically significant for the moth photograph task (Fig. S11B1), perhaps due to the limitations of this particular analysis outlined above. We can observe strong effects of image enhancement in both the moth photograph and dermoscopy tasks by examining the enhancement only (“Enhancement Taper/ET”) arm of the ablation study (Table 1 in the main text): for both tasks, participants in the ET condition had higher test-phase accuracy than participants in the control conditions, and these differences were statistically significant.
>
> “
>
> Furthermore, we have updated the analysis of image difficulty prediction in the appendix (please see Section S5 and Figure S4) to show more comprehensively that our human image difficulty prediction approach outperforms the prior state-of-the-art established by Mayo et al. 2023 [1].
>
> Finally, regarding image enhancement results in ImageNet, we have independently replicated the finding that human participants are significantly more accurate at classifying images enhanced by robust ANNs than the original, unmodified images, in 3 separate experiments with 3 sets of participants. The first is Figure 1, the second is the newly added Figure S6 in the Appendix (which compares enhancements from various ANN guide models), and the third is in Appendix Figure S8 (logit-maximization enhancement works on both validation set and training set images, and is superior to cross-entropy-minimization enhancement).
>
> References:
>
> [1] Mayo, David, Jesse Cummings, Xinyu Lin, Dan Gutfreund, Boris Katz, and Andrei Barbu. "How hard are computer vision datasets? Calibrating dataset difficulty to viewing time." Advances in Neural Information Processing Systems 36 (2023): 11008-11036.

---

> > ### Author Response · Authors · 2024-11-27
> > **Response to feedback from reviewer nJDg (part 2)**
> >
> > 2. _The related work section lacks discussion of both the machine teaching and human-machine vision alignment methods._
> >
> > We thank the reviewer for this comment. As suggested we have added a substantial Related Work section, including discussion of many additional works related to image enhancement, machine teaching and different ways of aligning models with human perception. Here is an excerpt relevant to this comment:
> >
> > “
> >
> >  Other studies focused on model-human alignment. Brain-Score directly benchmarks ANN mod- els with respect to neural representation and downstream behavior (Schrimpf et al., 2018); “Har- monization” methods directly drive alignment by an auxiliary objective on ANN-predicted feature importance maps and crowd-sourced ones (Fel et al., 2022). Other works introduce architecture components to account for additional aspects of human vision, such as the dorsal-stream “where” pathway in the brain (Choi et al., 2023).
> >
> > A key property that enables ANNs to generate human-interpretable image perturbations is that of perceptually aligned gradients, which is closely related to adversarial robustness and can be induced through adversarial training Ganz et al. (2023); Gaziv et al. (2024). Here, we apply adversarially-trained ANNs to enhance images such that they are more strongly associated with their ground truth label by the guiding model and by humans. To the best of our knowledge, we are the first to demonstrate improved human performance on image classification tasks through category-specific image enhancement.
> >
> > Our primary goal is to apply difficulty prediction and image enhancement to assist human learning. The emerging field of machine teaching (Zhu, 2015) employs machine learning to find or generate optimal “teaching sets” that can be used to train other models or humans. While many such approaches have been successfully applied to training machine learning models (e.g., Liu et al. (2017); Qiu et al. (2023)), few studies have successfully enhanced image category learning in humans and most of these focus on teaching set selection. Singla et al. (2014) propose STRICT, which optimizes the expected decrease in learner error based on how the selected images and their labels constrain a linear hypothesis class in a feature space. Johns et al. (2015) extend a similar approach to select images in an online fashion by modeling the learner’s progress. MaxGrad (Wang et al., 2021a) uses bi-level optimization to iteratively refine a teaching set by modeling learners as optimal empirical risk minimizers. Most similar to our work are approaches like EXPLAIN (Mac Aodha et al., 2018), which uses ANN class activation maps (CAMs) to highlight relevant image regions while providing feedback to the learner. EXPLAIN also selects a curriculum of images based on (a) a multi-class adaptation of STRICT, (b) representativeness (mean feature-space distance to other images of the same class), and (c) the estimated difficulty (entropy) of the CAM explanations. Chang et al. (2023) use bounding boxes to highlight image regions attended to by experts and not novices, allowing humans to more accurately match bird or flower images to one species among five shown in a gallery.
> >
> > “
> >
> >
> > 3. _The size of the participants is somewhat small._
> >
> > Our experiments involved more than 500 human participants with a minimum of 27 in each group for the learning experiments, and approximately 168 trials (learning experiments) or 352 trials (ImageNet recognition experiments) per participant. This is a high number of participants compared to most psychophysics papers. Our decisions regarding the number of participants was guided by a power analysis, which we have now included in Appendix Section S11. We report the results of statistical tests for all of the findings.
> >
> >
> > 4. _The perception of enhanced images may be altered due to perturbations._
> >
> > We agree with the reviewer that the perception of enhanced images is altered by perturbations: an important goal of our study was to alter perception through model-informed perturbations in such a way that enhances learning. We wonder if the reviewer considers this a weakness due to concerns about whether human learning from perturbed images generalizes to natural images - this is indeed a critically important consideration. Crucially, we **always** evaluate learning outcomes on **non-perturbed** images precisely for this reason. The fact that test-time accuracy (on natural, non-perturbed images) is higher for participants who saw perturbed images during training shows that altered perception during training provided generalizable benefits. We can observe this effect independently of image difficulty selection in the “Enhancement Taper (ET)” condition in the ablation study (Table 1).

---

> > > ### Author Response · Authors · 2024-11-27
> > > **Response to feedback from reviewer nJDg (part 3)**
> > >
> > > **Questions**
> > >
> > > 1. _The empirical observations are derived from a 16-way animal categorization task on natural images, which seems somewhat simplistic. It would be valuable to examine how these observations hold up in more complex categorization tasks or with different types of images, especially medical images. Given the typically limited availability of medical images, the proposed method could have promising applications in the medical imaging field._
> > >
> > > We thank the reviewer for this comment and agree that it is interesting to validate these empirical observations across multiple image domains, including complex categorization tasks such as those with medical images. To address this point, we have added Figure S11 to the Appendix, extending image difficulty prediction and image enhancement observations to the three fine-grained image category learning experiments. Paragraphs 2 and 3 of Appendix Section S8, which are newly added, read as follows:
> > >
> > > “
> > >
> > > In addition to the agreement of expert MHIST annotators, the ground truth logit successfully predicts the proportion of human participants who select the correct ground truth label across all tasks we tested. Difficulty prediction results from the 16-way ImageNet task are shown in Fig. 1 in the main text, and from the moth photograph, dermoscopy, and histology tasks in Fig. S11 (Panels A1, B1, and C). For the non-ImageNet tasks, we rely on test-phase data from control group participants who had just learned the tasks in question.
> > >
> > > We can also attempt to measure the extent to which images with higher levels of enhancement are easier for novice participants to recognize during the learning tasks (Fig. S11A2,B2). This analysis is limited to the first training trial blocks in the “ET (shuffled)” participant group in the ablation study (main-text Table 1), the only group that viewed enhanced images without monotonically decreasing ϵ. Note that there were 6 discrete ϵ values (1 per block in the non-shuffled ET condition), and the analysis is complicated by the fact that participants were still learning the task when they made the responses underlying these plots. We are also unable to compare with ϵ = 0 unmodified images because participants did not view new unmodified images in the corresponding training blocks. There are no results here for the histology task because the ablation study was conducted only for moth photos and dermoscopy images. In the dermoscopy task, participants respond with the original, correct category label statistically significantly more often when viewing images enhanced with greater ϵ (Fig. S11B2). A similar trend was not statistically significant for the moth photograph task (Fig. S11B1), perhaps due to the limitations of this particular analysis outlined above. We can observe strong effects of image enhancement in both the moth photograph and dermoscopy tasks by examining the enhancement only (“Enhancement Taper/ET”) arm of the ablation study (Table 1 in the main text): for both tasks, participants in the ET condition had higher test-phase accuracy than participants in the control conditions, and these differences were statistically significant.
> > >
> > > “

---

> ### Author Response · Authors · 2024-11-27
> **Response to feedback from reviewer nJDg (part 4)**
>
> 2. _Beyond the empirical observations, is there any physiological insight or analysis on why the proposed model-based selection and enhancement method could improve novice learners’ performance in categorization tasks?_
>
> We thank the reviewer for raising this question. It would be very interesting to (for example) conduct neurophysiological recordings while participants complete the learning tasks. Ultimately, however, the goal of this study was not to investigate the physiological mechanisms underlying visual perception or learning.
>
> The guiding principles behind designing L-WISE were not based on neurophysiology or neuroscience per se, but were rather established by behavioral observations from many past studies of human learning. Many studies have found that beginning a learning task at a low level of difficulty and then gradually increasing it while keeping the difficulty at a manageable level increases the speed of learning [1]. In particular, this type of ``curriculum effect’’ has been observed in perceptual learning of simple visual perceptual tasks, such as line orientation discrimination [2]: this led us to hypothesize that a similar effect could be observed with more complex and highly challenging visual perceptual learning tasks such as those involving medical images. L-WISE is designed to begin the learning process at a manageable difficulty level by selecting relatively easy images (based on image difficulty predictions) and enhancing them using models to reduce the difficulty further and highlight important features - it then gradually increases the difficulty by including more and more difficult images and enhancing them by perturbations limited to smaller and smaller $\epsilon$ values (see Figure 2, Panels E-F). We believe that this easy-to-hard curriculum effect at least partly explains L-WISE’s demonstrated efficacy at augmenting visual learning.
>
> References:
>
> [1]  Green, C. Shawn, and Daphne Bavelier. "Exercising your brain: a review of human brain plasticity and training-induced learning." Psychology and aging 23, no. 4 (2008): 692.
>
> [2] Lu, Zhong-Lin, and Barbara Anne Dosher. "Current directions in visual perceptual learning." Nature reviews psychology 1, no. 11 (2022): 654-668.
>
>
> 3. _The authors do not discuss the related machine teaching literature. An in-depth comparison with machine teaching methods, particularly with "Teaching Categories to Human Learners with Visual Explanations" (CVPR 2018), would be valuable. This work similarly considers image difficulty; an introduction and comparison with it are beneficial._
>
> We thank the reviewer for these suggestions - we were not aware of the suggested CVPR 2018 paper, which is indeed highly relevant. We have added a dedicated Related Work section to more comprehensively discuss prior works in machine teaching as well as difficulty prediction, image enhancement, and aligning ANN models with humans.
>
>
> 4. _The authors should also discuss the connection to human-machine vision alignment methods, such as "Harmonizing the Object Recognition Strategies of Deep Neural Networks with Humans" (NeurIPS 2022)_
>
> We thank the reviewer for this suggestion. We have included this paper and other papers that discuss human-machine vision alignment in our expanded Related Work section.
>
> 5. _The sample size of participants is relatively small, and expanding the participant pool is recommended to enhance the reliability of the results; also, recruiting participants from diverse backgrounds would improve the generalizability of the findings. If expanding the participant pool is impractical due to time or budget constraints, performing a power analysis or discussing effect sizes could help strengthen the reliability of the analysis._
>
> Our experiments involved more than 500 human participants with a minimum of 27 in each group for the learning experiments, and approximately 168 trials (learning experiments) or 352 trials (ImageNet recognition experiments) per participant. This is a high number of participants compared to most psychophysics papers. Our decisions regarding the number of participants was guided by a power analysis, which we have now included in Appendix Section S11. We report the results of statistical tests for all of the findings.
>
> We recruited participants who are diverse in age,  gender, and race/ethnicity (see new Table S2 in the Appendix).

---

> > ### Author Response · Authors · 2024-11-27
> > **Response to feedback from reviewer nJDg (part 5)**
> >
> > 6. _When using perturbations to enhance images, how does the method ensure that essential image details remain unchanged, particularly when using a large ϵ (e.g., 20)? This concern is especially pertinent for medical images, where even slight pixel changes may alter critical information. I recommend involving medical experts to review the enhanced images or using quantitative similarity measures (such as SSIM or FSIM) to verify that essential details are preserved._
> >
> > We understand this concern, and especially so in the case of medical image domains. Arguably, however, the ultimate test of any learning strategy is the accuracy performance on a held-out test set. Under this framing, in principle, if a method showing a handful of specialized random noise images to subjects for training resulted in a significant and consistent increase of test accuracy in a well-controlled study, we would have deemed it successful. In fact, this is sometimes successfully exercised in machine vision (e.g., Zada et al. ICML’22 [1]). Therefore, directly ensuring that all details are preserved in the enhanced images, used for training only, should not necessarily be required. Indeed, non-natural images are used extensively for medical education purposes: for example, hand-drawn illustrations have been successfully used to teach human anatomy in medical schools for centuries, and are still used extensively today even while high-resolution photographs and CT/MRI images are also used in the same courses [2][3]. Just as illustrations play a complementary role to natural images in anatomy instruction, model-enhanced images may one day play a complementary role in teaching visual tasks in medical domains such as histopathology.
> >
> > That said, we do indirectly attempt to preserve task-relevant information in the training images by limiting the pixel budget in the learning experiments to a maximum value of 8 – values of 20 are never used for these experiments. We have found that if perturbations are too strong, learners do reach high accuracy but then fail to generalize in the test phase (this is discussed in the first paragraph of Appendix Section S8: see Appendix Figure S9M for results of a pilot study that unsuccessfully used large-magnitude perturbations).
> >
> > We also have medical experts as part of our team, according to whom the perturbations to the medical images make sense and seem to accentuate task-relevant features. For example, a revised sentence in the Discussion section now reads: “Image enhancements might draw the learner’s attention to relevant features, such as…the multiple colors and irregular borders that appear to be enhanced in the melanoma
> > image of Fig. 4 (Tsao et al., 2015).”
> >
> > To better illustrate the nature of the perturbations for readers, we have added heat maps and “difference” images to each of the image examples in Figures 3 and 4 as well as Appendix Figure S1.
> >
> > References:
> >
> > [1] Zada, Shiran, Itay Benou, and Michal Irani. "Pure noise to the rescue of insufficient data: Improving imbalanced classification by training on random noise images." In International Conference on Machine Learning, pp. 25817-25833. PMLR, 2022.
> >
> > [2] Netter, Frank H. Netter Atlas of Human Anatomy: A Systems Approach-E-Book: Netter Atlas of Human Anatomy: A Systems Approach-E-Book. Elsevier Health Sciences, 2022.
> >
> > [3] Papa, Veronica, Elena Varotto, Mauro Vaccarezza, and Francesco M. Galassi. "Teaching Anatomy Through Images." Anthropologie (1962-) 59, no. 2 (2021): 145-154.

---

> > > ### Author Response · Authors · 2024-12-02
> > > **Gentle reminder regarding response to review**
> > >
> > > We thank the reviewer once again for their time in reading our paper and providing feedback. We understand that everybody is busy with reviewing papers and responding to reviewers as we approach today's deadline, but we would greatly appreciate any additional feedback on our revisions to the paper and the responses that we provided.

---

> > > > ### Comment · Reviewer_nJDg · 2024-12-03
> > > > **Response to the Rebuttal**
> > > >
> > > > Thank you for your response. The authors have addressed most of my concerns, and the quality of the manuscript has been further improved. Thus, I continue to maintain my score at 'marginally above the acceptance threshold'.

---

### Official Review · Reviewer_5U3k · 2024-11-04

**Soundness:** 3
**Presentation:** 3
**Contribution:** 2
**Rating:** 6
**Confidence:** 2

**Summary:**

This paper presents L-WISE, a framework leveraging adversarially robust ANNs to estimate image difficulty and apply nuanced perturbations that facilitate human learning in visual categorization tasks. By selecting challenging images and amplifying category-specific features, L-WISE improves human categorization accuracy by 33-72% and reduces training time by 20-23% across both general and clinical domains (e.g., dermoscopy and histology). The authors also discuss ethical implications, emphasizing the benefits of enhanced medical training and cautioning against potential biases that may arise from reliance on model-derived guidance.

**Strengths:**

1. The application of Robustified ANNs for improving human visual performance on image categorization seems like an interesting avenue.
2. L-Wise empirically demonstrate gains in categorization accuracy and training efficiency.
3. The paper addresses ethics concerns. Since the work mentions use of clinical data ethics discussion is of critical importance.

**Weaknesses:**

1. The paper focused on the performance of ventral stream. But we know that the human visual stream has a dorsal stream (where) that locates an object and the ventral stream (what) stream. And the interplay of these two streams forms the basis of human visual system. In this work the authors mainly focused on the ventral stream. From only quantified data, we can see the gains but it is very hard to trace this back to the nuanced perturbations the ANN produces. Hence, the suggestion is to use human gaze. The human gaze will precisely pin-point the "where" aspect and then will truly help us understand if at all the model perturbations are helping improve human performance. Can the authors please explain this?
2. A robust DNN actually has worse performance on nominal data points. Data points that have not been corrupted adversarially. What was the motivation of the authors to select such a model for their experiments?
3. The perturbations -  The perturbations if I am not mistaken are very subtle ones. For fine grained classifications, humans do follow curriculum learning but learning structures gradually from simpler to harder concepts. No experiments have shown this. It would be great if the authors can provide some empirical results/ explanation that can explain how will their method occur when you focus on structural cues rather than model perturbations that will benefit fine grained categorization.
4. I am providing some citations related to Gaze and dual stream hypothesis that can help authors clarify my concerns

a. A Dual-Stream Neural Network Explains the Functional Segregation of Dorsal and Ventral Visual Pathways in Human Brains, NeurIPS 2023.
b. Literature related to papers accepted in NeurIPS Gaze Meets ML workshop. That workshop accepted papers will provide intuition of how human gaze can be used in coherence with DL models.

**Questions:**

1. For model perturbations, can the authors please provide heat maps or any qualitative results that will help us track the ANN perturbations to human visual learning?
2. Is there a way to show if the study scales/generalizes to other ANNs as well other than Adversarially trained ones? Since human subjects have been used here, I am not sure how feasible experiments will be. This can be a general neural network or a network trained by the CutMix [a] loss that provides robustness benefits as well.
3. I feel generating perturbations based of dual stream networks and then using human gaze to track these will be a much stronger claim to the work. Can the authors please address this question?
4. What about adversarially trained transformers? The attention maps are different from CNN feature maps. How will the study be applicable for perturbations based of a transformer backbone?
5. Please also address concerns raised in the weakness sections.

a. CutMix: Regularization Strategy to Train Strong Classifiers with Localizable Features. In Proceedings of the IEEE/CVF International Conference on Computer Vision (ICCV), 2019,

---

> ### Author Response · Authors · 2024-11-27
> **Response to feedback from reviewer 5U3k (part 1)**
>
> We thank the reviewer for their time and effort in reviewing our submission, and for their insightful feedback, which has led to improvements in our manuscript.
>
> **Weaknesses**
>
> 1. _The paper focused on the performance of ventral stream. But we know that the human visual stream has a dorsal stream (where) that locates an object and the ventral stream (what) stream. And the interplay of these two streams forms the basis of human visual system. In this work the authors mainly focused on the ventral stream. From only quantified data, we can see the gains but it is very hard to trace this back to the nuanced perturbations the ANN produces. Hence, the suggestion is to use human gaze. The human gaze will precisely pin-point the "where" aspect and then will truly help us understand if at all the model perturbations are helping improve human performance. Can the authors please explain this?_
>
> We agree with the reviewer that a complete model of visual processing and object recognition behavior would have to account for dorsal stream processing and gaze behavior. We are unsure whether the reviewer is referring to the categorization or learning experiments. In the 16-way animal categorization task (Figure 1), images are presented for only 17 ms in the central portion of the participants’ vision, which does not leave enough time for the participant to make any eye movements. This is an established approach for studying ``_core_ object recognition’’ (recognition per saccade stations) in isolation from gaze-related effects [1].
>
> We agree with the reviewer that the “where” aspect of the image perturbations is important and interesting, and that it was not obvious to identify where exactly some of the images had been modified in the examples provided. To help clarify this and address the reviewer’s feedback, we have added “difference” images and heat maps showing which parts of the images are modified the most by the enhancement technique (please see Figures 3-4, and Appendix Figure S1). The changes are concentrated in semantically important regions of the images.
>
> We note that we can readily evaluate how model perturbations improve human performance. Our results demonstrate with a high degree of statistical reliability that category recognition accuracy is higher when images are enhanced by model perturbations in a task participants are already familiar with (Figure 1), and also that learners achieve higher accuracy on new, unfamiliar tasks, and in a shorter period of time, when they are trained with a curriculum that incorporates image enhancements (Figures 3 and 4). These effects have never been observed before, and we introduce these results as a proof-of-concept without claiming that the models we employed are optimal choices. As the reviewer suggests, a better model of human vision, such as one incorporating aspects of gaze or the dorsal stream, could potentially lead to an even more effective enhancement of image recognition and learning.
>
> References:
>
> [1] DiCarlo, James J., and David D. Cox. "Untangling invariant object recognition." Trends in cognitive sciences 11, no. 8 (2007): 333-341.
>
>
> 2. _A robust DNN actually has worse performance on nominal data points. Data points that have not been corrupted adversarially. What was the motivation of the authors to select such a model for their experiments?_
>
> The reviewer is correct that robust DNNs have worse performance on data points that have not been adversarially corrupted: this is referred to as the “robustness-accuracy tradeoff.” Our motivation for selecting robust models is based on existing evidence that robust models can produce perturbations that have strong disruptive effects on human category perception, while perturbations obtained in the same way with non-robust models have negligible or no impact [1].
>
> To quantify the effect of robustness on image enhancement, we have added Figure S6 to the Appendix - this figure shows that enhancement perturbations from robust models produce large improvements in image recognition accuracy, but perturbations of the same magnitude from non-robust (“vanilla”) models have no effect. We have also added Figure S7 to the Appendix which shows examples of image enhancements produced by robust and non-robust models, providing intuition on why this is the case. To quantify the impact of robustness on predicting the difficulty of images, we have expanded Figure S4 in the appendix. This figure shows that using robust models to predict the difficulty of each image for humans outperforms state-of-the-art approaches to predicting image difficulty using non-robust models.
>
> References:
>
> [1] Gaziv, Guy, Michael Lee, and James J. DiCarlo. "Strong and precise modulation of human percepts via robustified ANNs." Advances in Neural Information Processing Systems 36 (2024).

---

> ### Author Response · Authors · 2024-11-27
> **Response to feedback from reviewer 5U3k (part 2)**
>
> 3. _The perturbations - The perturbations if I am not mistaken are very subtle ones. For fine grained classifications, humans do follow curriculum learning but learning structures gradually from simpler to harder concepts. No experiments have shown this. It would be great if the authors can provide some empirical results/ explanation that can explain how will their method occur when you focus on structural cues rather than model perturbations that will benefit fine grained categorization._
>
> We agree with the reviewer about the importance of the curriculum order by which images are presented. That is why we designed L-WISE to progress from easy to hard examples, both by selecting easy images for the early part of the task (by predicting their difficulty with a robust model) and by decreasing the magnitude of enhancements as the task progresses (please see Figure 2, Panels E-F). We also explicitly study the effects of curriculum ordering in our ablation study of L-WISE (Table 1): regarding perturbations specifically, one can compare the performance of “enhancement taper (ET)” with “ET (shuffled)” - ET has a progression of more to less enhancement as the task progresses, while ET (shuffled) randomizes the order to erase this easy-to-hard trend. The fact that ET (shuffled) has lower performance than ET confirms that the decreasing enhancement schedule is helpful for learning.
>
> The image enhancement approach used in this study does not explicitly separate out specific visual concepts, but rather makes them more salient for the learner to notice - the degree to which each feature is enhanced by the model depends by construction on the model’s gradients, which are a reflection of which features are most important according to the model [1]. An interesting finding of our study is that allowing the model to decide which features to highlight has strong, measurable benefits for learners. Whether the L-WISE strategy overlaps with the notion of graded structural cues, or whether the latter is an effective strategy to confer a learning boost at all, is not a claimed contribution of the present study.
>
> References:
>
> [1] Selvaraju, Ramprasaath R., Michael Cogswell, Abhishek Das, Ramakrishna Vedantam, Devi Parikh, and Dhruv Batra. "Grad-CAM: visual explanations from deep networks via gradient-based localization." International journal of computer vision 128 (2020): 336-359.
>
> 4. _I am providing some citations related to Gaze and dual stream hypothesis that can help authors clarify my concerns_
>
> We thank the reviewer for suggesting this very interesting paper (which we have now cited in our related work section), and we think it is a great suggestion to incorporate aspects of human gaze in conjunction with deep learning models. We are currently planning a study of human gaze and models incorporating it in the context of medical images, although that work is not ready for inclusion in the present manuscript.
>
> We wish to emphasize that our current work is a proof-of-concept showing that, with use of a model that incorporates (even incompletely) some aspects of human visual perception, we can strongly improve visual recognition accuracy and learning performance by enhancing images and predicting their difficulty. This is an entirely novel finding. It is quite likely that the results could be improved even further with a more comprehensive model of the human visual system, and the reviewer’s suggestion of incorporating dual-stream neural networks is insightful and interesting.
>
>
> **Questions**
>
> 1. _For model perturbations, can the authors please provide heat maps or any qualitative results that will help us track the ANN perturbations to human visual learning?_
>
> Inspired by this suggestion, we have added heat maps and ``difference’’ images to figures 3-4, and to the newly added Appendix Figure S1. We believe this improves the interpretability aspect of the work: it can be observed qualitatively that the ANN perturbations are localized in semantically important regions of the images.

---

> > ### Author Response · Authors · 2024-11-27
> > **Response to feedback from reviewer 5U3k (part 3)**
> >
> > 2. _Is there a way to show if the study scales/generalizes to other ANNs as well other than Adversarially trained ones? Since human subjects have been used here, I am not sure how feasible experiments will be. This can be a general neural network or a network trained by the CutMix [a] loss that provides robustness benefits as well._
> >
> > This is another great suggestion, which we have addressed with an entirely new experiment (involving more than 60 additional human participants) that compares the accuracy of difficulty predictions and the effects of perturbations across 6 different ANN models, including a more generally-used ANN (ResNet-50 without adversarial training), and ResNet-50 trained with CutMix per the reviewer’s suggestions. Please see Figures S5 and S6 in the appendix for quantitative results, and Appendix Figure S7 for examples of images perturbed using the different guide models. Consistent with prior work [1], we find that using robust models is necessary to produce perturbations that have strong effects on human visual perception. Interestingly, although the reviewer is correct that CutMix confers a degree of robustness to adversarial perturbations, this was not sufficient to induce a capability for strong perceptual modulation or difficulty prediction. The fact that our technique works with adversarially trained/robustified models and not other models is not inherently a limitation of our work: rather, it highlights the usefulness of robustified models specifically in certain application domains.
> >
> > References:
> >
> > [1] Gaziv, Guy, Michael Lee, and James J. DiCarlo. "Strong and precise modulation of human percepts via robustified ANNs." Advances in Neural Information Processing Systems 36 (2024).
> >
> > 3. _I feel generating perturbations based of dual stream networks and then using human gaze to track these will be a much stronger claim to the work. Can the authors please address this question?_
> >
> > We appreciate the reviewer’s insight that using dual-stream networks and tracking human gaze could lead to even stronger results - indeed, we are planning future experiments that track gaze. We wish to emphasize that, although the models of visual perception we have used in this study could be improved upon in various ways (including by modeling aspects of the dorsal stream), they are _sufficient_ to demonstrate substantial improvements in human image recognition performance and in assisting human visual category learning. These are new findings with high potential for practical applications in education, and future works could improve upon the methods developed here by using dual-stream networks and tracking human gaze. We think the exciting part of our work is showing that enhancing perception and learning is possible in this manner in the first place.
> >
> > 4. _What about adversarially trained transformers? The attention maps are different from CNN feature maps. How will the study be applicable for perturbations based of a transformer backbone?_
> >
> > We agree with the reviewer that it is interesting to test different adversarially trained ANN architectures, particularly vision transformers given their recent success and widespread use. In this spirit, we conducted an entirely new experiment involving 60 human participants to compare the image enhancement and difficulty prediction capabilities of 6 different ANNs (see also our response to question 2 from this reviewer). Among them is robustified XCiT [1], a vision transformer architecture that is better suited to adversarial pretraining than the original ViT [2]. The perturbations from robustified XCiT improved accuracy comparably to robustified ResNet-50 (Appendix Figure S6), and difficulty prediction accuracy was also comparable to ResNet-50 (Appendix Figure S5).
> >
> > References:
> >
> > [1] Ali, Alaaeldin, Hugo Touvron, Mathilde Caron, Piotr Bojanowski, Matthijs Douze, Armand Joulin, Ivan Laptev et al. "Xcit: Cross-covariance image transformers." Advances in neural information processing systems 34 (2021): 20014-20027.
> >
> > [2] Debenedetti, Edoardo, Vikash Sehwag, and Prateek Mittal. "A light recipe to train robust vision transformers." In 2023 IEEE Conference on Secure and Trustworthy Machine Learning (SaTML), pp. 225-253. IEEE, 2023.
> >
> > 5. _Please also address concerns raised in the weakness sections._
> >
> > We believe we have addressed all concerns in the weakness sections.
> >
> >
> > **Ethics**
> >
> > _Flag For Ethics Review: Yes, Responsible research practice (e.g., human subjects, data release)_
> >
> > The involvement of human participants requires careful attention to ethics and best practices. We have added some additional information regarding our practices surrounding human subjects research and data in the first paragraph of the Ethics Statement (Section 7).

---

> > > ### Comment · Reviewer_5U3k · 2024-12-01
> > > **Response to the rebuttal**
> > >
> > > > Thank you for the detailed and comprehensive response addressing my concerns and comments. Additionally, the authors have improved the clarity of the figures in the main manuscript and added relevant sections to the Appendix. Based on these improvements, I have updated my overall rating and assessment of the work. The revised manuscript is much improved and reads well.
> > >
> > > Strengths of the Revision:
> > >
> > > My broad concerns regarding the following aspects have been addressed comprehensively:
> > >
> > >  - The use of a robust trained model as the candidate surrogate model for the study (Weakness
> > >    [2]).
> > >  - The incorporation of perturbations as part of a curriculum
> > >    learning strategy (Weakness [3]).
> > >  - Related work and its contextual relevance (Weakness [4]).
> > >  - Specific questions related to these points    (Questions [2], [3],
> > >    [4]).
> > >
> > > > The justification for using a robustly trained model as the surrogate model strengthens the contributions of the work. Furthermore, the discussion on applying the approach to other network architectures and learning paradigms enhances the generalizability and impact of the method.
> > >
> > > I have one remaining question, which pertains to the heatmaps, and I would appreciate clarification if I am misunderstanding.
> > >
> > > **Clarification on Heatmap Generation and Interpretation**
> > >
> > > I have a question regarding the generation and interpretation of the heatmaps:
> > >
> > > -   **Heatmap Generation Method:** Are these heatmaps generated based on a model's predictive outputs, or do they strictly represent changes in the images due to perturbations without involving a model in their creation? In other words, were any predictive modeling techniques used to generate these heatmaps, or do they simply visualize the differences caused by the perturbations in the images themselves?
> > >
> > > -   **Interpretation of Brighter Regions:** Do the brighter regions in the heatmaps reflect where the model focuses its attention to make predictions based on the most salient parts of the image, or do they indicate areas where the perturbations have affected the image regardless of any model's influence?
> > >
> > > I understand the perturbations are generated by a surrogate model. My question strictly concerns the generation process of the heatmaps.
> > >
> > > **Impact on Human Learning:**
> > >
> > > -   To verify if these perturbations improve human learning for categorization, it seems critical to check if the perturbed regions align with the central part of a participant's vision.
> > > -   My understanding is that the effectiveness of perturbations in aiding human learning depends on whether these perturbed regions coincide with the participant's gaze. If they do not, then the perturbations may not contribute significantly to the learning process.
> > > -   Furthermore, the temporal aspect of 17 ms seems crucial here. To allow participants' gaze to focus on different perturbed regions, this duration might need to be increased.
> > >
> > > If the authors could clarify whether my understanding is accurate and address these points, it would be greatly appreciated.

---

> > > > ### Author Response · Authors · 2024-12-01
> > > > **Response to reviewer's follow-up questions (part 1)**
> > > >
> > > > We  thank the reviewer for their time in reading the revised manuscript and our responses, and for their encouraging words acknowledging the improvements. We respond to the reviewer’s additional questions below.
> > > >
> > > > **Clarification on Heatmap Generation and Interpretation**
> > > >
> > > >
> > > > 1. _**Heatmap Generation Method:** Are these heatmaps generated based on a model's predictive outputs, or do they strictly represent changes in the images due to perturbations without involving a model in their creation? In other words, were any predictive modeling techniques used to generate these heatmaps, or do they simply visualize the differences caused by the perturbations in the images themselves?_
> > > >
> > > > This is an important clarification to address and we thank the reviewer for highlighting it: we will show all steps to generate the heatmaps explicitly in an additional portion of Section S1 of the camera-ready version. The reviewer’s latter suggestion is completely correct: ``the heatmaps simply visualize the differences caused by the perturbations in the images themselves.’’ We generate the heatmaps by subtracting enhanced image $x’$ from original image $x$ element-wise: $\delta = x’ - x$. We calculate the magnitude of the changes as $\delta^2$. We apply smoothing using 2D convolution, and normalize the result to have all values between 0 and 1, to produce $\delta_{\text{norm}}$. We then produce the heat map by setting the red channel to $255 \times \delta_{\text{norm}}$ and the blue channel to $255 \times (1 - \delta_{\text{norm}})$. The result shows red in regions where larger changes have taken place, and blue in regions where smaller or no changes have taken place.
> > > >
> > > >
> > > > 2. _**Interpretation of Brighter Regions:** Do the brighter regions in the heatmaps reflect where the model focuses its attention to make predictions based on the most salient parts of the image, or do they indicate areas where the perturbations have affected the image regardless of any model's influence?_
> > > >
> > > >
> > > > _I understand the perturbations are generated by a surrogate model. My question strictly concerns the generation process of the heatmaps._
> > > >
> > > > As noted above, the heatmaps show differences between the model-enhanced images and the original images. The model generates perturbations based on predictions of what will make the image easier to recognize: intuitively, the model makes the image easier for itself to recognize, which also makes it easier for humans to recognize in the case of models with perceptually aligned gradients such as robustified models. The model does not try to make predictions about bottom-up saliency, top-down saliency, etc. However, salient regions in an image (as defined by, say, bottom-up saliency) could correlate with areas that the model decides to enhance or not.
> > > >
> > > >
> > > >
> > > >
> > > > **Impact on Human Learning:**
> > > >
> > > >
> > > > 3. _To verify if these perturbations improve human learning for categorization, it seems critical to check if the perturbed regions align with the central part of a participant's vision._
> > > >
> > > >
> > > > Whether or not the participants directly foveate the perturbed regions is an interesting question. In the majority of the images, the target object encompasses the central part of the image, which correlates with the central part of a participant’s vision (although we did not record any eye movements). In the camera-ready version, we will compute a map averaging across all images showing the main regions that are changed by the model. However, whether or not participants foveate these regions, we can still measure the degree of improvement in learning outcomes in terms of categorization accuracy: assessing foveation is not required to demonstrate this, nor is foveation a confounding factor.
> > > >
> > > >
> > > > We wish to emphasize that our experiments do verify extensively that the perturbations empirically improve human learning: **in the participant groups with L-WISE assistance, accuracy on non-perturbed images during a final test phase is higher than in control participants by factors of 33-72%** – these are statistically significant differences, and the finding is robust across multiple image domains. We also study enhancement in isolation in the ablation study (Table 1), where it has large, statistically significant benefits relative to controls (see ``Enhancement Taper (ET).’’

---

> > > > > ### Author Response · Authors · 2024-12-01
> > > > > **Response to reviewer's follow-up questions (part 2)**
> > > > >
> > > > > 4. _My understanding is that the effectiveness of perturbations in aiding human learning depends on whether these perturbed regions coincide with the participant's gaze. If they do not, then the perturbations may not contribute significantly to the learning process._
> > > > >
> > > > >
> > > > > We operationally define learning efficacy based on recognition performance on unmodified, held-out test images. We have demonstrated that the image perturbations do contribute significantly to the learning process: this is a statistically significant finding that we have replicated across multiple image domains. We agree with reviewers that it is interesting to assess whether the changed regions correlate with fixation locations. It is also conceivable that participants could benefit from the enhancements even if they fixate away from the image changes. The idea of using image perturbations to aid human learning is entirely new in our study: indeed, category-specific image enhancement itself is a novel contribution of our paper (as explained in the new Related Work section). This opens up an entirely new sub-field of which one paper can only scratch the surface: examining how category-specific image enhancements by ANNs affect human category perception, visual attention, gaze, aesthetic judgments, various forms of visual learning, and other factors. Gaze is not the focus of the present paper. We are, however, particularly interested in the role of gaze and visual search: we are working on a future project that applies the methods pioneered by the present paper in the context of a visual search task, where gaze becomes especially important.
> > > > >
> > > > >
> > > > >
> > > > >
> > > > > 5. _Furthermore, the temporal aspect of 17 ms seems crucial here. To allow participants' gaze to focus on different perturbed regions, this duration might need to be increased._
> > > > >
> > > > >
> > > > > The reviewer is correct that 17ms is not sufficient for the participants’ gaze to focus on different regions. This 17ms presentation time applies specifically to our experiments with ImageNet (see Figure 1 and Appendix Figures S1, S4-S8), where we employ established methodology surrounding “core object recognition” [1]. In this sub-field, the images are presented only to the central, high-acuity portion of the participant’s vision (making eye movements unnecessary to capture the whole image in high acuity), and for extremely short durations of 200ms or less. In studies of core object recognition, this is actually intended to prevent eye movements, and such short stimulus durations are widely used (e.g., [1, 2, 3, 4, 5]). The reviewer is correct that results are likely to depend on image presentation time. In the specific case of our experiment, short presentation times also serve to make the task more challenging: otherwise, participants would be near 100% accuracy at distinguishing dogs from crabs from lizards etc. - which would make it impossible to quantify the effects of enhancement on a familiar object recognition task if the control group is already at the performance ceiling.
> > > > >
> > > > >
> > > > > Unlike for the ImageNet task, our image category learning experiments allow participants to view images for up to 10 seconds before making a response - during which many eye movements almost certainly occur. In this setting also, we show strong, positive effects of image perturbations (see Table 1 ``Enhancement Taper (ET)’’ condition, and Appendix Figure S11). **Therefore, we have demonstrated that the image enhancements are effective in both a setting where eye movements are not possible (ImageNet) and also in a setting where many eye movements are possible (learning tasks).**
> > > > >
> > > > >
> > > > > References:
> > > > >
> > > > >
> > > > > [1] DiCarlo, James J., and David D. Cox. "Untangling invariant object recognition." Trends in cognitive sciences 11, no. 8 (2007): 333-341.
> > > > >
> > > > >
> > > > > [2] Zhang, Mengmi, Claire Tseng, and Gabriel Kreiman. "Putting visual object recognition in context." In Proceedings of the IEEE/CVF conference on computer vision and pattern recognition, pp. 12985-12994. 2020.
> > > > >
> > > > >
> > > > > [3] Tang, Hanlin, Martin Schrimpf, William Lotter, Charlotte Moerman, Ana Paredes, Josue Ortega Caro, Walter Hardesty, David Cox, and Gabriel Kreiman. "Recurrent computations for visual pattern completion." Proceedings of the National Academy of Sciences 115, no. 35 (2018): 8835-8840.
> > > > >
> > > > >
> > > > > [4] Gaziv, Guy, Michael Lee, and James J. DiCarlo. "Strong and precise modulation of human percepts via robustified ANNs." Advances in Neural Information Processing Systems 36 (2024).
> > > > >
> > > > >
> > > > > [5] Kar, Kohitij, Jonas Kubilius, Kailyn Schmidt, Elias B. Issa, and James J. DiCarlo. "Evidence that recurrent circuits are critical to the ventral stream’s execution of core object recognition behavior." Nature neuroscience 22, no. 6 (2019): 974-983.

---

### Official Review · Reviewer_wUUM · 2024-11-04

**Soundness:** 1
**Presentation:** 2
**Contribution:** 3
**Rating:** 6
**Confidence:** 4

**Summary:**

The paper proposes a novel approach to augment human learning in image categorization tasks. By leveraging robustified ANNs, the study introduces model-guided image selection and enhancement strategies that increase human test-time categorization accuracy by up to 72% and reduce training duration by around 20-23%. L-WISE employs selecting images based on predicted difficulty levels and enhancing images with pixel perturbations.The proposed approach is tested on natural images, dermoscopy, and histology images. The results demonstrates efficacy of L-WISE in aiding novice learners in fine-grained categorization tasks. This research represents one of the first applications of ANNs in optimizing human visual learning in clinically relevant domains.

**Strengths:**

- Presents an innovative use of robustified ANNs to predict task difficulty and enhance images, aiding human perception and learning.
- Shows broad applicability by successfully testing across diverse domains, such as natural image classification, dermoscopy, and histology.
- Achieves practical efficiency by reducing training time and improving test-time accuracy, beneficial for fields requiring rapid, accurate human image categorization training.

**Weaknesses:**

- Lacks a dedicated related work section, which would help contextualize the research.
- Both low and high logits from ANNs show significant variation in human accuracy, making predictions less reliable in certain logit intervals.
- Uses only the ResNet-50 architecture, limiting generalization; further testing with models like vision transformers (ViT) is needed to support broader conclusions.
- Image enhancement may introduce biases, potentially improving accuracy only for certain major classes; additional metrics like precision and recall per class, rather than just mean accuracy, should be reported to provide a clearer assessment.

**Questions:**

1. How to choose $\epsilon$ for different tasks and image domains?
2. What criteria determine if a model is "robustified" enough for use? Have you considered specific metrics to evaluate the robustness of guide models, and how do these metrics correlate with human learning outcomes?
3. Did you collect qualitative feedbacks from participants? Did the new curriculum and enhanced images increase mental stress of human learners? Additional learning costs beyond training time should be considered, such as cognitive load and emotional well-being.

**Details Of Ethics Concerns:**

The use of ANN-guided models in human learning may introduce unintended biases, potentially affecting participants' learning outcomes in ways that favor certain demographic groups over others. For instance, if the training of ANNs inadvertently enhances accuracy for only a subset of the population (such as specific genders, ages, or races), it could lead to biased learning outcomes. Such biases could potentially skew related job opportunities, ultimately reinforcing inequities. Ensuring that ANNs benefit all demographic groups equitably requires further investigation and ongoing evaluation to mitigate these risks.

The above concern is addressed.

---

> ### Author Response · Authors · 2024-11-27
> **Response to feedback from reviewer wUUM (part 1)**
>
> We thank the reviewer for their time and effort in reviewing our submission, and for their insightful feedback, which has led to improvements in our manuscript.
>
> **Weaknesses**
>
> 1. _Lacks a dedicated related work section, which would help contextualize the research._
>
> We agree with the reviewer, and as suggested, we have now added a substantial Related Work section (Section 2), including discussion of many additional works related to image enhancement, machine teaching, and different ways of aligning models with human perception.
>
>
> 2. _Both low and high logits from ANNs show significant variation in human accuracy, making predictions less reliable in certain logit intervals._
>
> As indicated by the reviewer, we agree that the 95% confidence intervals are wider at very low and very high logit values. This, however, is not necessarily due to less reliable predictions but rather reflects the limited statistical power of our measurement in those logit ranges. There are fewer images with extreme logit values as the images are roughly normally distributed along this axis. To increase the statistical power even in the extreme logit value ranges, we have updated Panel A of Figure 1 to incorporate newly collected data from additional human participants, resulting in narrower confidence intervals throughout the logit range.
>
> We also wish to emphasize that our image difficulty prediction approach outperforms the prior state-of-the-art in this regime by a considerable margin. In addition to the change to Figure 1, we have updated the analysis in the appendix (please see updated Section S5 and new Figure S4) to more comprehensively compare our human image difficulty prediction method with the prior state-of-the-art established by Mayo et al. 2023 [1]. In the newly expanded analysis, we demonstrate that the ground truth logit of a robust model is a significantly better predictor than the c-score [2] (approximated by the epoch during training at which an image is first correctly predicted), image-level adversarial robustness [1] (perturbation size required to change the model’s prediction of the image), and prediction depth [3] (earliest layer upon which a linear probe makes the same prediction as the final output) calculated using both vanilla and robustified models.
>
> References:
>
> [1] Mayo, David, Jesse Cummings, Xinyu Lin, Dan Gutfreund, Boris Katz, and Andrei Barbu. "How hard are computer vision datasets? Calibrating dataset difficulty to viewing time." Advances in Neural Information Processing Systems 36 (2023): 11008-11036.
>
> [2] Jiang, Ziheng, Chiyuan Zhang, Kunal Talwar, and Michael C. Mozer. "Characterizing Structural Regularities of Labeled Data in Overparameterized Models." In International Conference on Machine Learning, pp. 5034-5044. PMLR, 2021.
>
> [3] Baldock, Robert, Hartmut Maennel, and Behnam Neyshabur. "Deep learning through the lens of example difficulty." Advances in Neural Information Processing Systems 34 (2021): 10876-10889.

---

> ### Author Response · Authors · 2024-11-27
> **Response to feedback from reviewer wUUM (part 2)**
>
> 3. _Uses only the ResNet-50 architecture, limiting generalization; further testing with models like vision transformers (ViT) is needed to support broader conclusions._
>
> This work shows that a reasonable model of human vision, which is thought to be among the leading models of the ventral visual stream, can be used to enhance image category learning in humans; we do **not** claim that human learning can be augmented using **any** ANN architecture, nor do we claim that robustified ResNet-50 is optimal for this goal. Our work is rather a proof-of-concept showing that augmenting human learning with an ANN is possible, and across varied and potentially useful image domains. To the best of our knowledge, this is the first work demonstrating the possibility of enhancing visual learning in humans by using any computational model to enhance images.
>
> The choice of ANN architecture is one of many hyperparameters in L-WISE that could potentially be further optimized - others include the strength of image perturbations, the temporal profile of enhancement tapering and difficulty selection, the overall length of the curricula, and the data used for pretraining the ANN guide models. Hyperparameter tuning in this setting is time and capital-intensive given that dozens to hundreds of human participants must be recruited for each experiment. In our present work, we are focused on establishing proof-of-concept and leave further engineering optimizations for future works.
>
> However, we agree with the reviewer that it is interesting to test different ANN architectures, particularly vision transformers given their recent success and widespread use. In this spirit, we conducted an entirely new experiment involving 60 human participants to compare the image enhancement and difficulty prediction capabilities of 6 different ANNs. Among them is robustified XCiT [1], a vision transformer architecture that is better suited to adversarial pretraining than the original ViT [2]. The perturbations from robustified XCiT improved accuracy comparably to robustified ResNet-50 (See Appendix Figure S6), and difficulty prediction accuracy is also comparable between the two models (Appendix Figure S5).
>
> References:
>
> [1] Ali, Alaaeldin, Hugo Touvron, Mathilde Caron, Piotr Bojanowski, Matthijs Douze, Armand Joulin, Ivan Laptev et al. "Xcit: Cross-covariance image transformers." Advances in neural information processing systems 34 (2021): 20014-20027.
>
> [2] Debenedetti, Edoardo, Vikash Sehwag, and Prateek Mittal. "A light recipe to train robust vision transformers." In 2023 IEEE Conference on Secure and Trustworthy Machine Learning (SaTML), pp. 225-253. IEEE, 2023.
>
> 4. _Image enhancement may introduce biases, potentially improving accuracy only for certain major classes; additional metrics like precision and recall per class, rather than just mean accuracy, should be reported to provide a clearer assessment._
>
> The possibility of L-WISE improving recognition of some classes and not others is an important concern that was not addressed in our initial submission. We have added Panel B to Figure 4 (and Table S1 to the Appendix) showing that, while precision and recall do vary among classes, both precision and recall are improved by L-WISE for each class across all three image domains we tested. The reviewer’s intuition was correct that improvements to precision and recall, while universally positive, do appear larger in magnitude for some classes than others.

---

> > ### Author Response · Authors · 2024-11-27
> > **Response to feedback from reviewer wUUM (part 3)**
> >
> > **Questions**
> >
> > 1. _How to choose $\epsilon$ for different tasks and image domains?_
> >
> > The choice of $\epsilon$ for different tasks and image domains is one of many interesting hyperparameters involved in our approach. In our approach to assist learning, we first show to the learner images perturbed with a maximal $\epsilon$ value and then decrease this according to a fixed schedule - it would be interesting to vary both the starting $\epsilon$ and the schedule profile. As our present work is intended as a proof-of-concept, we did not optimize either of these extensively - we find that the exact same $\epsilon$ profile is effective at enhancing human learning across all three of the image domains we tested (as discussed in Section 6/Methods, see “Assisting learners with the L-WISE algorithm”).
> >
> > Nevertheless, some simple conclusions can be drawn about optimal $\epsilon$ values from the data we have collected. Although larger perturbations may further increase recognition accuracy (see Figure 1, Panel B1), the results of one of our pilot experiments (which involved only enhancement tapering and not difficulty-based selection) suggested that excessively large perturbations such as $\epsilon=20$ can hinder the learner’s ability to generalize their knowledge to non-perturbed images (please see Figure S9M in the Appendix). After this pilot experiment, we limited the maximal magnitude of perturbations in L-WISE to $\epsilon=8$, which was sufficient to facilitate generalizable learning across image domains.
> >
> > We also conducted an ablation study to evaluate the importance of $\epsilon$ decreasing as the task progresses (see Table 1). In the table, enhancement tapering (“ET”) is isolated from the full L-WISE algorithm (i.e., no difficulty prediction-based sequencing), and compared with the further ablated “ET (shuffled),” in which the $\epsilon$ fluctuates randomly throughout training instead of gradually decreasing (because the order of the images in the training phase are shuffled after having been enhanced according to the original, decreasing schedule). ET (shuffled) is inferior to ET for both moth photograph and dermoscopy image learning, suggesting that decreasing $\epsilon$ as learning progresses is helpful for learning.
> >
> > 2. _What criteria determine if a model is "robustified" enough for use? Have you considered specific metrics to evaluate the robustness of guide models, and how do these metrics correlate with human learning outcomes?_
> >
> > Our key goal in this study was to evaluate the _feasibility_ of using models to enhance perception and learning. We are not claiming that the proposed models or other settings of our approach are optimal in any sense and it is quite possible that models with different levels of robustness could be more effective.
> >
> > As suggested by the reviewer, the degree of robustification is indeed an important determinant of the guide model’s effectiveness. The robustness of the guide model can be adjusted by changing the $\epsilon$ magnitude of perturbations generated during adversarial pretraining. In a new experiment added for this rebuttal, we compared $\epsilon=1$, $\epsilon=3$, and $\epsilon=10$ pretrained ResNet-50 models in terms of their ability to predict image difficulty and enhance image category perception (see Figures S5 and S6 in the appendix). Consistent with the disruption efficiency findings of [1], we find that $\epsilon=3$ is superior to $\epsilon=1$ and $\epsilon=10$ in both difficulty prediction and enhancement perturbation. Additionally, we have added the following clarification to Section S1 in the Appendix:
> > “Our choice of $\epsilon$=3 for ImageNet pretraining follows Gaziv et al (2024), who found this to be an optimal choice for perturbations that disrupt category perception (relative to $\epsilon$=1 and $\epsilon=10$). In practice, we found that the models were unable to learn finer-grained tasks with training-time adversarial perturbations as large as $\epsilon$=3 (iNaturalist pretraining and fine-tuning on moth photos, dermoscopy images, and histology images) - therefore, we reverted to $\epsilon$=1 for these settings.”
> >
> > References:
> >
> > [1] Gaziv, Guy, Michael Lee, and James J. DiCarlo. "Strong and precise modulation of human percepts via robustified ANNs." Advances in Neural Information Processing Systems 36 (2024).

---

> ### Author Response · Authors · 2024-11-27
> **Response to feedback from reviewer wUUM (part 4)**
>
> 3. _Did you collect qualitative feedbacks from participants? Did the new curriculum and enhanced images increase mental stress of human learners? Additional learning costs beyond training time should be considered, such as cognitive load and emotional well-being._
>
> We agree with the reviewer that factors such as mental/emotional stress and cognitive load are crucially important, particularly if techniques similar to L-WISE are used in educational settings in the future. The present study was focused on proof-of-concept rather than practical applications, and as such we did not formally collect qualitative feedback. In future work that is more oriented towards classroom/clinical learning, we will systematically collect qualitative feedback - this is an excellent suggestion.
>
> We predict that the effects of L-WISE on stress and cognitive load in an educational setting would actually be positive, not negative. We were motivated to create L-WISE by observations that certain visual tasks, particularly in the clinical realm, are very difficult for beginners and thus frustrating to learn at the early stages. L-WISE ameliorates this by selecting easier images for the beginning of the learning process, and also enhancing them to make it easier to recognize important features. Many studies on human learning have found that beginning a learning task at a low level of difficulty and then gradually increasing it while keeping the difficulty at a manageable level, increases both the speed of learning and the motivation of learners [1]. One thing we can measure in our experiments is how often participants withdraw from the experiment before completing the whole learning task, a common occurrence in online studies where humans attempt to learn difficult tasks [2]. We present a new analysis of task withdrawal rates by condition in Appendix Section S9. Participants who were assisted by L-WISE were significantly more likely (p < 0.05) to complete the entire learning task, suggesting that they found the task more interesting and/or less frustrating than participants in the control group without L-WISE assistance.
>
> References:
>
> [1]  Green, C. Shawn, and Daphne Bavelier. "Exercising your brain: a review of human brain plasticity and training-induced learning." Psychology and aging 23, no. 4 (2008): 692.
>
> [2] Wang, Pei, Kabir Nagrecha, and Nuno Vasconcelos. "Gradient-based algorithms for machine teaching." In Proceedings of the IEEE/CVF Conference on Computer Vision and Pattern Recognition, pp. 1387-1396. 2021.
>
> **Ethics**
>
> 1. _The use of ANN-guided models in human learning may introduce unintended biases, potentially affecting participants' learning outcomes in ways that favor certain demographic groups over others. For instance, if the training of ANNs inadvertently enhances accuracy for only a subset of the population (such as specific genders, ages, or races), it could lead to biased learning outcomes. Such biases could potentially skew related job opportunities, ultimately reinforcing inequities. Ensuring that ANNs benefit all demographic groups equitably requires further investigation and ongoing evaluation to mitigate these risks._
>
> We thank the reviewer for raising this concern. Our recruited sample of participants was diverse in age, gender, and race/ethnicity - we have added a table of participant demographics to the Appendix (Table S2). In general, it is indeed a concerning possibility that AI-assisted learning could disproportionately benefit certain population groups due to the models capturing group-specific characteristics of visual processing. We argue that this type of bias in learning assistance is highly unlikely for the types of visual tasks we focus on. There is no evidence of strong or fundamental differences in basic visual perception or learning among different genders or ethnic groups [1], with the possible exception of face recognition (which is not the focus of this study) [2]. L-WISE works by making challenging tasks easier at the beginning and then gradually increasing the difficulty - it would be quite surprising if this was fundamentally helpful for some demographic groups but not others. This study demonstrates that boosting human learning via model-generated image perturbations, previously unknown to be possible at all, is achievable and in a diverse sample of participants. Future studies will be needed to evaluate potential biases such as those raised here.
>
> References:
>
> [1] Kreiman, Gabriel. Biological and computer vision. Cambridge University Press, 2021.
>
> [2] Young, Steven G., Kurt Hugenberg, Michael J. Bernstein, and Donald F. Sacco. "Perception and motivation in face recognition: A critical review of theories of the cross-race effect." Personality and Social Psychology Review 16, no. 2 (2012): 116-142.

---

> > ### Author Response · Authors · 2024-12-02
> > **Gentle reminder regarding response to review**
> >
> > We thank the reviewer once again for their time in reading our paper and providing feedback. We understand that everybody is busy with reviewing papers and responding to reviewers as we approach today's deadline, but we would greatly appreciate any additional feedback on our revisions to the paper and the responses that we provided.

---

> > > ### Comment · Reviewer_wUUM · 2024-12-03
> > > **Response to the Rebuttal.**
> > >
> > > Thank you for your response. I acknowledge that most of my concerns are addressed in the rebuttal. Hence, I adjust my score to " marginally above the acceptance threshold".

---

### Author Response · Authors · 2024-11-27
**Summary of changes in rebuttal revision**

We thank the reviewers for all of their thoughtful feedback and insights, and for their encouraging words expressing interest in our work. We agree that the constructive feedback provided would improve the manuscript and clarify its contributions. Below, we provide a summary of the most important changes implemented in this revision:

**Related Work:**

In response to reviewer feedback we have added a substantial dedicated Related Work section, with discussion on prior approaches to image difficulty prediction, image enhancement, alignment of ANN models with humans, and using ANN models to assist human learning (machine teaching).

**Figures & Presentation:**

**Fig. 1:**
- Added data points from additional human participants to logit predictions (panel A) to reduce measurement uncertainty.
- Revised example image selection.
- New, cleaner figure design (for example, example images in panel B2 are now lined up directly underneath the corresponding data in panel B1).
- Fixed/added text labels.
- Added example images to provide intuition for the proposed model-based recognition difficulty score.

**Fig. 2:**
- Cleaner and clearer explanatory text labels.
- Fixed typos.
- Changed focus from robustified models to ANN-based classification models more broadly.

**Fig. 3:**
- Added interpretability images: heatmaps and ``difference’’ images indicating which image regions are affected by enhancement.
- Cleaner/clearer figure design and text labels.

**Fig. 4:**
- New figure design.
- Improved visualization of the joint test-accuracy and training speed gains as a scatter plot.
- New analysis for per–class precision and recall metrics across the three considered image domains.
- Added interpretability images for dermoscopy and histology domains: heatmaps and ``difference’’ images between original and enhanced image versions.
- Cleaner and clearer explanatory text labels.

**Added/Extended Supplementary Material:**

**Fig. S1** - Additional examples of ImageNet image enhancement, with interpretability images similar to the new Figures 3 and 4.

**Fig. S4** - Expanded analysis that more comprehensively compares the proposed image difficulty score with the prior state of the art.

**Fig. S5** - Extension of image difficulty prediction to other model types and robustification levels (robustified vision transformer, CutMix, ResNet-50 non-robustified and robustified using $\epsilon=1,3,10$).

**Fig. S6** - Quantitative human psychophysics analysis of image category enhancement effectiveness by various models (robustified vision transformer, CutMix, ResNet50 non-robustified and robustified using $\epsilon=1,3,10$).

**Fig. S7** - Qualitative visual analysis of enhancement effects by various models (robustified vision transformer, CutMix, ResNet50 non-robustified and robustified using $\epsilon=1,3,10$).

**Fig. S11** - Support for empirical observations presented (for the 16-way ImageNet task) in Fig 1 for the three different image domains focused on in the learning tasks.

**Table S1** - Precision/Recall per class in every learning task considered, including confidence intervals.

**Table S2** - Demographic characteristics of study participants, aggregated across all experiments.

**Section S6** - Comparing different guide models (robustified vision transformer, CutMix, ResNet-50 non-robustified and robustified using $\epsilon=1,3,10$).

**Section S8** - Paragraphs 2-4 are new, outlining extension of image difficulty prediction and image enhancement observations to learning domains and class-wise breakdown of learning gains attributed to L-WISE.

**Section S9** - New analysis showing that participants who receive L-WISE assistance are more likely than control group participants to persist through the entire learning task.

**Section S11** - Details on participant recruitment and demographics.

---

### Meta-Review · Area_Chair_tk6T · 2024-12-20

**Metareview:**

This paper presents an advancement in using ANNs to enhance human visual learning of image categories. The research makes several good contributions to both theoretical understanding and practical applications.
The primary contribution is the development of L-WISE (Logit-Weighted Image Selection and Enhancement) which uses robustified ANNs to improve how humans learn to categorize images. The system works in two ways: it predicts how difficult images will be for humans to categorise, and it can enhance images to make their key features more recognisable. When combined, these capabilities allow for creating optimised learning sequences that significantly improve both learning speed and accuracy.
A key strength of the research is its comprehensive empirical validation. The authors demonstrated L-WISE's effectiveness across multiple domains, including natural images (moth species), medical dermoscopy (skin lesions), and histology (tissue samples). The results were good, showing improvements in accuracy and training time.

The research also makes a theoretical contribution by establishing that robustified ANNs can serve as accurate predictors of image recognition difficulty for humans. This finding extends our understanding of human-AI alignment and suggests new ways that AI models can be used to understand and enhance human perception.

However, the study does have some limitations. The authors acknowledge they didn't exhaustively explore optimal curriculum design strategies or enhancement parameters. There's also a key concern about the homogenising effect of their enhancement technique - for example, enhanced images of benign moles tend to look very similar, potentially obscuring important real-world variations that medical practitioners need to recognise.

Despite these limitations, the research represents a good step forward in using AI to enhance human learning. The fact that the improvements were consistent across different domains and tasks suggests the approach has broad applicability. Furthermore, the authors' transparent discussion of limitations and potential risks demonstrates a thoughtful consideration of the wider implications of their work.

**Additional Comments On Reviewer Discussion:**

Overall the reviewers have been positive and have unanimously voted to accept this paper, albeit marginally. One reviewer (5U3k) indicated that they would be increasing their score to 8 but this never happened.
Some issues were raised around the limitations of the method, choice of architecture, and ethics/bias, which led to some thoughtful responses by the authors.
Given the confidence of the reviewers, I believe this paper is marginal.

---

### Decision · Program_Chairs · 2025-01-22

Accept (Poster)